PREPARED FOR SUBMISSION TO JHEP

# Bulk locality from the celestial amplitude

**Chi-Ming Chang**[1,2] **Yu-tin Huang**[3,4] **Zi-Xun Huang**[3] **Wei Li**[3]

[1] *Yau Mathematical Sciences Center (YMSC), Tsinghua University, Beijing, 100084, China*

[2] *Beijing Institute of Mathematical Sciences and Applications (BIMSA), Beijing, 101408, China*

[3] *Department of Physics and Center for Theoretical Physics, National Taiwan University, Taipei, Taiwan 106*

[4] *Physics Division, National Center for Theoretical Sciences, Taipei 10617, Taiwan*

*E-mail:* cmchang@tsinghua.edu.cn, yutinyt@gmail.com, r09222060@ntu.edu.tw, r07222072@ntu.edu.tw

ABSTRACT: In this paper, we study the implications of bulk locality on the celestial amplitude. In the context of the four-point amplitude, the fact that the bulk S-matrix factorizes locally in poles of Mandelstam variables is reflected in the imaginary part of the celestial amplitude. In particular, on the positive real axes in the complex plane of the boost weight, the imaginary part of the celestial amplitude can be given as a positive expansion on the Poincaré partial waves, which are nothing but the projection of flat-space spinning polynomials onto the celestial sphere. Furthermore, we derive the celestial dispersion relation, which relates the imaginary part to the residue of the celestial amplitude for negative even integer boost weight. The latter is precisely the projection of low energy EFT coefficients onto the celestial sphere. We demonstrate these properties explicitly on the open and closed string celestial amplitudes. Finally, we give an explicit expansion of the Poincaré partial waves in terms of 2D conformal partial waves.

# 1   Introduction

In recent years, there has been steady progress on understanding the 2D holographic description of 4D flat-space scattering amplitudes [1–21] introduced by the pioneering work [1, 4], where one replaces the asymptotic state of the scattering amplitudes with boost eigenstates. In particular, the action of the Lorentz group $SL(2, \mathbb{C})$ on the kinematic data is recast into the Möbius transform on the celestial sphere, and the scattering amplitude is reinterpreted as a correlation function for some two-dimensional conformal field theory (CFT), termed the celestial amplitude. For amplitudes of massless external particles, this change of basis is implemented by a Mellin transform and the quantum numbers of the primary operator $(h, \bar{h})$ is related to the helicity $(\ell)$ of the external particle as $\ell = h - \bar{h}$, while the dimension $\Delta = h + \bar{h}$ is in-principle unconstrained.[1]

As the celestial amplitudes (denoted as $\tilde{\mathcal{A}}_n$ for $n$ states) are defined on boost eigenstates, which superpose all energies, the usual Wilsonian decoupling of UV/IR physics no long applies and is only well-defined for theories equipped with a UV completion. This combined with the lack of local observables in quantum gravity, makes the celestial amplitude the prime arena to study general properties of consistent quantum gravity theories [21]. Motivated by this, it will be desirable to derive the general set of consistency conditions for $\tilde{\mathcal{A}}$.

The analytic properties of flat-space amplitude is an intensely studied subject and is relatively well-understood within the realm of perturbation theory. Thus, the "projection" of these properties onto the celestial sphere should serve as the primary constraint. For massless amplitudes, one of the simplest universal behaviors are the soft limits. However, due to the superposition of all energies, for $\tilde{\mathcal{A}}_n$ the fate of soft theorems were unclear. The first progress toward elucidating the image of flat-space soft theorems were taken in [14, 15, 22], where the limit was realized in the limit where the conformal dimensions are taken to 1. More precisely, these "conformal soft limits" of the celestial amplitudes lead to various conformal Ward identities associated with the holomorphic currents that generate the Kac-Moody symmetry in gauge theory [15], or the BMS supertranslation and the Virasoro symmetries in gravity [16, 19]. The conformal soft theorems then constrain the leading operator product expansion (OPE) coefficients in the celestial CFT [18].

On the other hand, the actions of the Poincaré symmetry on the celestial sphere are explicitly worked out in [12], and their constraints on the four and lower point celestial amplitudes are investigated in detailed in [17]. As a consequence a new set of expansion basis for $\tilde{\mathcal{A}}_4$, the Poincaré (relativistic) partial waves, was proposed in [20] analogous to the conformal partial wave expansion of four-point functions in CFT.

The above progresses focuses on the various symmetry properties of the celestial amplitudes and the constraints followed from them. An obvious gap is the role of flat-space factorization, which encodes *locality*, through where singularity might occur, and unitary that governs the residue or discontinuity across the singularity. Indeed it is often the case that these considerations alone are sufficient to completely determine the flat-space

---

[1]Completeness relation of the conformal partial waves requires $\Delta = 1 + i\mathbb{R}$.

amplitude. In view of this one would like to ask:

*How does the bulk-locality reflect on celestial amplitudes?*

Limit analysis was done for three-dimensional scalar exchange [6], as well as four-dimensional massive external legs [2]. In this paper, we aim to address this question in general for the four-point massless celestial amplitudes.

Poincaré invariance fixes the form of $\tilde{\mathcal{A}}_4$ up to a function that depends only on the real conformal cross ratio $z = \bar{z} = \frac{z_{12}z_{34}}{z_{13}z_{24}}$, the total conformal dimension (boost weight) $\beta = \Delta_1 + \Delta_2 + \Delta_3 + \Delta_4 - 4$, and the helicities $\ell_i$, for $i = 1, \cdots, 4$ of the external particles [12, 17]. The resulting function $\Psi(\beta, \ell_i, z)$ is related to the scattering amplitude in the plane wave basis by a Mellin transform

$$\Psi(\beta, \ell_i, z) \propto \int_0^\infty d\omega\, \omega^{\beta-1} T_{\ell_i}(s, t) \ , \tag{1.1}$$

where $\omega$ is the center of mass energy and $T_{\ell_i}(s, t)$ is the flat-space amplitude stripped off the momentum conservation delta function and a kinematic phase. For fixed $\{\ell_i\}$ the we have a function of two variables $(\beta, z)$, replacing the flat-space parameterization $(s, t)$. It was argued in [21] that the function is analytic in $\beta$ except for integer values on the real axes. The poles for $\beta = -2\mathbb{Z}^+$ are controlled by the Wilson coefficients of the low energy EFT with the degree of the poles determined by the IR running. For $\beta = 2\mathbb{Z}^+$, these encodes the polynomial suppression of the UV amplitude. For theories of quantum gravity that latter singularities are expected to be absent due to black hole productions.

Note that while the four-point celestial amplitude is defined on the equator of the celestial sphere, its not a continuous function across the entire circle. Using the usual $SL(2,\mathbb{C})$ to fix three points to $(0, 1, \infty)$, the equator is divided into three regions corresponding to $s, t, u$-channel kinematics respectively. Each channel is named after the positive Mandelstam variable, while the remaining two are negative in the physical region. Schematically, we have:

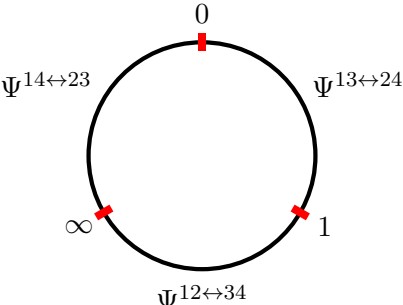

where the superscript on $\Psi$ denotes the physical channel. Importantly, due to the $\beta$ dependence of the Mellin transform, there are non-trivial monodromies across the three branch points $(0, 1, \infty)$, even for $\beta = 2\mathbb{Z}$. Thus, the three functions are in fact distinct and are not analytically connected, as first observed in the context of three-dimensions [6]. We will first demonstrate that due to bulk factorization, each function will acquire an imaginary piece reflecting the presence of thresholds in the physical channel, i.e. the channel with positive center of mass energy. In particular, the imaginary part of the celestial amplitude

can be computed by extending the original Mellin integration to a fan-like contour that enclosed the positive real axis:

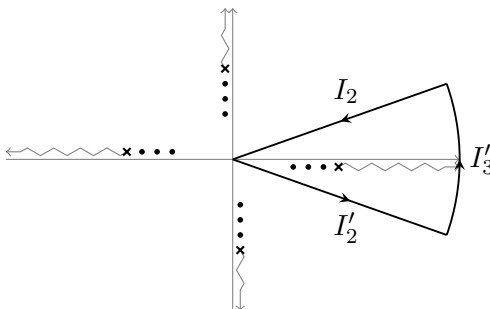

The contour integral captures the poles and discontinuities of the amplitude which can be expanded in the basis of orthogonal polynomials, Legendre polynomials for external scalars and Jacobi polynomials for gauge boson or graviton amplitudes. The projection of these polynomials onto the celestial sphere are then nothing but the Poincaré partial waves $\Phi_{m,J}(\beta, \ell_i, z)$ of mass $m$ and spin $J$ introduced in [20]. Thus, factorization singularities of the flat-space amplitude, in the physical channel, is projected into the imaginary part of the celestial amplitude and given by a sum of Poincaré partial waves, schematically,

$$\mathbf{Im}\,\Psi(\beta, \ell_i, z) = \sum_a p_a \Phi_{m_a, J_a}(\beta, \ell_i, z)\,. \tag{1.2}$$

If the external states are organized such that for the physical threshold corresponds to $a, b \to b, a$ process, then we further have $p_a > 0$, a reflection of unitarity.

Recently, it was shown that the EFT coefficients are constrained through dispersion relations in a fashion that reflects an underlying positive geometry, the EFThedron [23]. Since as previously mentioned, for $\tilde{\mathcal{A}}_4$ the poles on the negative $\beta$ axes encodes the EFT coefficients, these must be expressible as some form of dispersion relation. However, while the usual flat-space dispersion relation involves imaginary pieces arising from thresholds in distinct channels, for the celestial amplitude the imaginary part at any point on the equator is given by thresholds in one channel along. To this end we analytic continue the celestial amplitudes outside their physical defining regions to the "unphysical" regions. This allows us to establish the *celestial dispersion relation*, given as:

$$\frac{\pi}{2} \operatorname*{\mathbf{Res}}_{\beta \to -2n} \left[ \Psi^{12 \leftrightarrow 34}(\beta, z) \right] = \mathbf{Im} \left[ \Psi^{12 \leftrightarrow 34}(\beta, z) + (-1)^n \Psi^{13 \leftrightarrow 24}(\beta, z) \right] \Big|_{\beta \to -2n} \quad (z \geq 1)\,, \tag{1.3}$$

where we've given the form in the region $z \geq 1$. Note that the RHS contains the imaginary part of both $s$ and $u$-channel functions, where each can be defined in terms of the fan-like contour integral. We've explicitly verified these results for massive scalar exchange, as well as open and closed string amplitudes.

This paper is organized as follows. Section 2 reviews the kinematics of the four-point massless celestial amplitudes and the UV/IR behaviors onward. Section 3 computes explicitly the imaginary parts of the four-point massless celestial amplitudes, show their positive

expansion in terms of the Poincaré partial waves, and check the crossing symmetry. Section 4 studies the example of the imaginary part of the open and closed string celestial amplitudes. Section 5 discusses the analytic continuation of the celestial amplitudes. Section 6 derives the celestial dispersion relation and verifies it for the example of the open and closed string amplitudes. Section 7 ends with a summary, further comments and future directions. Appendix A derives the general form of helicity amplitudes that satisfy the constraints from the Lorentz symmetry and momentum conservation. Appendix B reviews the action of the Poincaré generators on a massless single particle state. Appendix D expands the celestial amplitude and the Poincaré partial waves in terms of the conformal partial waves. Appendix E computes the three-point structure constant that shows up in the conformal partial wave expansion.

## 2 Review of the four-point celestial amplitude

In this section, we review general properties of the four-point celestial amplitude. We will focus on massless amplitudes $\mathcal{A}$, where the momenta are given as $p_i = \epsilon_i \omega_i q_i^\mu$, with $\epsilon = \pm$ denoting whether the particle is outgoing or incoming, and the null vector $q^\mu$ is parametrized by

$$q_i^\mu = (1 + |z_i|^2, 2\mathrm{Re}(z_i), 2\mathrm{Im}(z_i), 1 - |z_i|^2). \tag{2.1}$$

Later on, $z_i$ will be the complex stereographic coordinate on the celestial sphere. Thus the amplitude $\mathcal{A}$ instead of being a function of four momenta, is now a function of $(\omega_i, z_i)$. The celestial amplitude is then simply the Mellin transform of helicity amplitudes:

$$\tilde{\mathcal{A}}_{\Delta_i, \ell_i}(z_i, \bar{z}_i) = \Big(\prod_{i=1}^n \int_0^\infty \mathrm{d}\omega_i \omega_i^{\Delta_i - 1}\Big) \mathcal{A}_{\ell_i}(\omega_i, z_i), \tag{2.2}$$

where $\ell_i$ is the helicity of each leg.

### 2.1 Space-time to Celestial sphere "kinematics"

Let us consider in detail the transformation of flat-space scattering amplitudes to the celestial sphere. Since we will be interested in helicity amplitudes, it is natural to embed $(\omega_i, z_i)$ in the spinor variables $\epsilon_i \omega_i q_i^\mu(\sigma_\mu) = \lambda_i \tilde{\lambda}_i$, where:

$$\lambda_i = \epsilon_i \sqrt{2\omega_i} \begin{pmatrix} 1 \\ z_i \end{pmatrix}, \quad \tilde{\lambda}_i = \sqrt{2\omega_i} \begin{pmatrix} 1 \\ \bar{z}_i \end{pmatrix}. \tag{2.3}$$

The map is of course not unique, as any U(1) rotation $\lambda \to e^{i\theta}\lambda$ and $\tilde{\lambda} \to e^{-i\theta}\tilde{\lambda}$ preserves the same null vector. The fact that the first component of $\lambda$ ($\tilde{\lambda}$) is real, correspond to a specific choice of frame. In this "canonical frame", the Lorentz invariant spinor brackets take the form:

$$\langle ij \rangle = \varepsilon^{ab} \lambda_{j,a} \lambda_{i,b} = 2\epsilon_i \epsilon_j \sqrt{\omega_i \omega_j}(z_i - z_j), \quad [ij] = \varepsilon^{\dot{a}\dot{b}} \tilde{\lambda}_{j,\dot{a}} \tilde{\lambda}_{i,\dot{b}} = 2\sqrt{\omega_i \omega_j}(\bar{z}_i - \bar{z}_j). \tag{2.4}$$

Similarly the Mandelstam variable $s_{ij} = \langle ij \rangle [ji]$ is related to the distance $|z_{ij}|$ on the celestial sphere by

$$s_{ij} = -(p_i + p_j)^2 = -2p_i \cdot p_j = 4\epsilon_i \epsilon_j \omega_i \omega_j |z_{ij}|^2 \,. \tag{2.5}$$

One can straight forwardly see that the $SL(2,\mathbb{C})$ Lorentz transformation acting on the spinors translate to the Möbius transformation acting on the complex plane $z$:

$$\begin{pmatrix} a & b \\ c & d \end{pmatrix} \lambda = e^{i\theta} \lambda', \quad z' = \frac{c+dz}{a+bz}, \quad \omega' = \omega |a+bz|^2, \quad e^{i\theta} = \frac{a+bz}{|a+bz|} \,. \tag{2.6}$$

Importantly, while the spinor products are Lorentz invariants, when considered in terms of the celestial coordinates they acquire a "little group" phase under $SL(2,\mathbb{C})$:

$$\begin{aligned}
\langle ij \rangle' &= 2\epsilon_i \epsilon_j \sqrt{\omega_i' \omega_j'} (z_i' - z_j') = 2e^{-i\theta_i} e^{-i\theta_j} \epsilon_i \epsilon_j \sqrt{\omega_i \omega_j} (z_i - z_j) \,, \\
[ij]' &= 2\sqrt{\omega_i' \omega_j'} (\bar{z}_i' - \bar{z}_j') = 2e^{i\theta_i} e^{i\theta_j} \sqrt{\omega_i \omega_j} (\bar{z}_i - \bar{z}_j) \,.
\end{aligned} \tag{2.7}$$

The origin of the extra phase is simple: as seen in eq.(2.3) an arbitrary $SL(2,\mathbb{C})$ transformation will rotate the spinors out of the canonical frame. Thus one requires a "compensating transformation" to restore it back. Since the amplitude transforms covariantly under little group transformations, manifested as helicity weights for each leg, one immediately deduce that the amplitude transform under $SL(2,\mathbb{C})$ as:

$$\mathcal{A}_{\ell_i}(\omega_i', z_i', \bar{z}_i') = \Big( \prod_j e^{2\ell_j \theta_j} \Big) \mathcal{A}_{\ell_i}(\omega_i, z_i, \bar{z}_i) \,. \tag{2.8}$$

The Mellin-transform in eq.(2.2) can then be viewed as changing from plane wave basis, to the conformal primary basis (or boost) [1, 4]. As a result under SL(2,C) transformation $\tilde{\mathcal{A}}_{\Delta_i, J_i}$ transforms as:

$$\tilde{\mathcal{A}}_{\Delta_i, \ell_i}(z_i', \bar{z}_i') = \Big( \prod_j (a+bz_j)^{\Delta_j + \ell_j} (\bar{a} + \bar{b}\bar{z}_j)^{\Delta_j - \ell_j} \Big) \tilde{\mathcal{A}}_{\Delta_i, \ell_j}(z_i, \bar{z}_i) \,. \tag{2.9}$$

Thus the $n$-point celestial amplitude $\tilde{\mathcal{A}}$ transforms like a $n$-point conformal correlator in two-dimensional conformal field theory

$$\langle O_{h_1, \bar{h}_1}(z_1, \bar{z}_1) \dots O_{h_n, \bar{h}_n}(z_n, \bar{z}_n) \rangle \,, \tag{2.10}$$

where the left-moving and right-moving conformal dimensions $h_i$ and $\bar{h}_i$ are

$$h_i + \bar{h}_i = \Delta_i, \quad h_i - \bar{h}_i = \ell_i \,. \tag{2.11}$$

The variables $\Delta_i$ in the Mellin transform (2.2) and the helicities $\ell_i$ become the total scaling dimensions and spins.

In this paper, our main focus is on the 4-point amplitude. $SL(2,\mathbb{C})$ conformal symmetry constrains the amplitude to the form

$$\tilde{\mathcal{A}}_{\Delta_i, \ell_i}(z_i, \bar{z}_i) = \frac{\left(\frac{z_{14}}{z_{13}}\right)^{h_3 - h_4} \left(\frac{z_{24}}{z_{14}}\right)^{h_1 - h_2} \left(\frac{\bar{z}_{14}}{\bar{z}_{13}}\right)^{\bar{h}_3 - \bar{h}_4} \left(\frac{\bar{z}_{24}}{\bar{z}_{14}}\right)^{\bar{h}_1 - \bar{h}_2}}{z_{12}^{h_1 + h_2} z_{34}^{h_3 + h_4} \bar{z}_{12}^{\bar{h}_1 + \bar{h}_2} \bar{z}_{34}^{\bar{h}_3 + \bar{h}_4}} f_{\Delta_i, \ell_i}(z, \bar{z}) \,, \tag{2.12}$$

where the cross ratios $z$ and $\bar{z}$ and coordinate differences $z_{ij}$, $\bar{z}_{ij}$ are

$$z = \frac{z_{12}z_{34}}{z_{13}z_{24}} \quad, \bar{z} = \frac{\bar{z}_{12}\bar{z}_{34}}{\bar{z}_{13}\bar{z}_{24}}, \quad z_{ij} = z_i - z_j, \quad \bar{z}_{ij} = \bar{z}_i - \bar{z}_j. \tag{2.13}$$

Translation invariance, or momentum conservation, further constrain the dependence on the cross-ratio [12, 17],

$$f_{\Delta_i,\ell_i}(z, \bar{z}) = (z-1)^{\frac{\Delta_1-\Delta_2-\Delta_3+\Delta_4}{2}}\delta(iz-i\bar{z})\Psi(\boldsymbol{\Delta}, \ell_i, z) \tag{2.14}$$

where $\boldsymbol{\Delta} = \sum_i \Delta_i$. Due to the delta function $\delta(iz - i\bar{z})$, the celestial amplitude to be supported on the equator of the celestial sphere.

One can also derive (2.12) and (2.14) directly from the Mellin integral representation (2.2). First, as show in Appendix A by utilizing the momentum conservation and the $\mathrm{SL}(2,\mathbb{C})$ symmetry, the four-point helicity amplitude take the form as

$$\mathcal{A}_{\ell_i}(\omega_i, z_i) = \delta^{(4)}(p_1 + p_2 + p_3 + p_4)\frac{\left(\frac{z_{14}\bar{z}_{13}}{\bar{z}_{14}z_{13}}\right)^{\frac{\ell_3-\ell_4}{2}}\left(\frac{z_{24}\bar{z}_{14}}{\bar{z}_{24}z_{14}}\right)^{\frac{\ell_1-\ell_2}{2}}}{\left(\frac{z_{12}}{\bar{z}_{12}}\right)^{\frac{\ell_1+\ell_2}{2}}\left(\frac{z_{34}}{\bar{z}_{34}}\right)^{\frac{\ell_3+\ell_4}{2}}}T_{\ell_i}(s, t), \tag{2.15}$$

where $s \equiv s_{12}$, $t = s_{14}$, and $u = s_{13}$ are the Mandelstam variables. The momentum conservation written in terms of the energies $\omega_i$ and the celestial sphere coordinates $z_i$ and $\bar{z}_i$ as

$$\delta\left(\sum_{i=1}^{4}\epsilon_i\omega_i q_i\right) = \frac{8}{\Lambda^2|\omega_4|}\delta(iz-i\bar{z})\delta\left(\omega_1+\epsilon_1\epsilon_4\frac{\Lambda^2\omega_4}{z}\right)\delta\left(\omega_2+\epsilon_2\epsilon_4\frac{\Lambda^2\omega_4}{z(z-1)}\right)\delta\left(\omega_3+\epsilon_3\epsilon_4\frac{\Lambda^2\omega_4}{1-z}\right), \tag{2.16}$$

where we have used the $\mathrm{SL}(2,\mathbb{C})$ transformation to fix the coordinates $z_i$ to

$$z_1 = 0, \quad z_2 = z, \quad z_3 = 1, \quad z_4 = \Lambda \gg 1, \tag{2.17}$$

In this conformal frame, the celestial scalar amplitude becomes

$$\begin{aligned}\tilde{\mathcal{A}}_{\Delta_i,\ell_i}(z_i, \bar{z}_i) = {}& 8\Lambda^{-2\Delta_4}|z-1|^{\frac{1}{2}(\Delta_1-\Delta_2-\Delta_3+\Delta_4)}|z|^{-\Delta_1-\Delta_2}\delta(iz-i\bar{z})\\ &\times \theta(-\epsilon_1\epsilon_4 z)\theta(\epsilon_2\epsilon_4 z(1-z))\theta(\epsilon_3\epsilon_4(z-1))\\ &\times z^2|z-1|^{2-\frac{\boldsymbol{\Delta}}{2}}\int_0^\infty \mathrm{d}\tilde{\omega}_4\,\tilde{\omega}_4^{\boldsymbol{\Delta}-5}T_{\ell_i}\left(\frac{4\tilde{\omega}_4^2}{z-1}, -\frac{4\tilde{\omega}_4^2}{z}\right),\end{aligned} \tag{2.18}$$

where $\tilde{\omega}_4 = \Lambda^{-2}\omega_4$. Note that in the conformal frame (2.17) the $z$-dependent factor in front of $T(s, t)$ in (2.15) reduces to 1 by the delta functions. The Heaviside theta functions are there to ensuring the delta functions having support in the integration domain of $\omega_i$. Indeed (2.18) reduces to (2.12) and (2.14) in the conformal frame (2.17) where one reads off $\Psi(\boldsymbol{\Delta}, \ell_i, z)$ as

$$\begin{aligned}\Psi(\boldsymbol{\Delta}, \ell_i, z) = {}& \theta(-\epsilon_1\epsilon_4 z)\theta(\epsilon_2\epsilon_4 z(1-z))\theta(\epsilon_3\epsilon_4(z-1))\\ &\times z^2|z-1|^{2-\frac{\boldsymbol{\Delta}}{2}}\int_0^\infty \mathrm{d}\tilde{\omega}_4\,\tilde{\omega}_4^{\boldsymbol{\Delta}-5}T_{\ell_i}\left(\frac{4\tilde{\omega}_4^2}{z-1}, -\frac{4\tilde{\omega}_4^2}{z}\right).\end{aligned} \tag{2.19}$$

| kinematics | $12 \leftrightarrow 34$ | $13 \leftrightarrow 24$ | $14 \leftrightarrow 23$ |
|---|---|---|---|
| physical region | $z \geq 1$ <br> $s \geq 0 \geq u, t$ | $1 \geq z \geq 0$ <br> $u \geq 0 \geq s, t$ | $0 \geq z$ <br> $t \geq 0 \geq s, u$ |
| $\omega$ | $\frac{2\tilde{\omega}_4}{\sqrt{z-1}}$ | $\frac{2\tilde{\omega}_4}{\sqrt{z(1-z)}}$ | $\frac{2\tilde{\omega}_4}{\sqrt{-z}}$ |
| $(s, u, t)$ | $\left(\omega^2, -\frac{1}{z}\omega^2, -\frac{(z-1)}{z}\omega^2\right)$ | $\left(-z\omega^2, \omega^2, -(1-z)\omega^2\right)$ | $\left(-\frac{(-z)}{1-z}\omega^2, -\frac{1}{1-z}\omega^2, \omega^2\right)$ |

**Table 1**. The physical regions, center of mass energy $\omega$ and Mandelstam variables in the three different kinematics.

Now due to the step functions, depending on the choice of incoming legs the cross-ratio $z$ is constrained to different regions. Consider three distinct kinematic configuration distinguished by the incoming state being in $s, u$ or $t$-channel,

$$
\begin{aligned}
12 \leftrightarrow 34 : \quad & \epsilon_1 = \epsilon_2 = -\epsilon_3 = -\epsilon_4 \,, \\
13 \leftrightarrow 24 : \quad & \epsilon_1 = \epsilon_3 = -\epsilon_2 = -\epsilon_4 \,, \\
14 \leftrightarrow 23 : \quad & \epsilon_1 = \epsilon_4 = -\epsilon_2 = -\epsilon_3 \,.
\end{aligned}
\tag{2.20}
$$

The Heaviside theta functions in (2.19) will constrain the celestial amplitudes with the three different kinematics to have supports on three separate intervals on the equator of the celestial sphere,

$$
12 \leftrightarrow 34 : z \geq 1 \,, \quad 13 \leftrightarrow 24 : 0 \leq z \leq 1, \quad 14 \leftrightarrow 23 : z \leq 0 \,.
\tag{2.21}
$$

It would be convenient to use the center of mass energy $\omega$ as the integration variable. We apply the changes of variables in the three different kinematics as

$$
12 \leftrightarrow 34 : \ \omega^2 = \frac{4\tilde{\omega}_4^2}{z-1} \,, \quad 13 \leftrightarrow 24 : \ \omega^2 = \frac{4\tilde{\omega}_4^2}{z(1-z)} \,, \quad 14 \leftrightarrow 23 : \ \omega^2 = \frac{4\tilde{\omega}_4^2}{(-z)} \,.
\tag{2.22}
$$

The physical regions, changes of variables, and the corresponding parametrizations of the Mandelstam variables are summarized in Table 1. Note that we can also identify $z = \frac{2}{1-\cos\theta}$, where $\theta$ is the scattering angle and the limit $z \to \infty$ corresponds to the forward limit.

The celestial amplitude (2.19) in these three kinematics are then defined as:

$$
\begin{aligned}
\Psi^{12 \leftrightarrow 34}(\boldsymbol{\Delta}, \ell_i, z) &= \frac{1}{2^{\boldsymbol{\Delta}-7}} z^2 \int_0^\infty d\omega\, \omega^{\boldsymbol{\Delta}-5} T_{\ell_i}\left(\omega^2, -\frac{(z-1)}{z}\omega^2\right) & (z \geq 1)\,, \\
\Psi^{13 \leftrightarrow 24}(\boldsymbol{\Delta}, \ell_i, z) &= \frac{1}{2^{\boldsymbol{\Delta}-7}} z^{\frac{\boldsymbol{\Delta}}{2}} \int_0^\infty d\omega\, \omega^{\boldsymbol{\Delta}-5} T_{\ell_i}\left(-z\omega^2, (z-1)\omega^2\right) & (1 \geq z \geq 0)\,, \\
\Psi^{14 \leftrightarrow 23}(\boldsymbol{\Delta}, \ell_i, z) &= \frac{1}{2^{\boldsymbol{\Delta}-7}} (-z)^{\frac{\boldsymbol{\Delta}}{2}} (1-z)^{2-\frac{\boldsymbol{\Delta}}{2}} \int_0^\infty d\omega\, \omega^{\boldsymbol{\Delta}-5} T_{\ell_i}\left(\frac{z}{1-z}\omega^2, \omega^2\right) & (0 \geq z)\,.
\end{aligned}
\tag{2.23}
$$

We stress that the celestial amplitude *is not* given by a single function $\Psi(\boldsymbol{\Delta}, z)$ defined on the equator. Rather, there are three separate functions $\Psi^{12 \leftrightarrow 34}$, $\Psi^{13 \leftrightarrow 24}$ and $\Psi^{14 \leftrightarrow 23}$ that tile the equator.

In short, the function $\Psi^{ij\leftrightarrow kl}(\boldsymbol{\Delta}, \ell_i, z)$ will be related to the amplitude via

$$\Psi^{ij\leftrightarrow kl}(\beta, \ell_i, z) = B^{ij\leftrightarrow kl}(z) \int_0^\infty d\omega\, \omega^{\beta-1} T_{\ell_i}^{ij\leftrightarrow kl}(\omega, z)\,, \qquad (2.24)$$

where using the notation of [21] we introduce $\beta = \boldsymbol{\Delta}-4$. $B^{ij\leftrightarrow kl}(z)$ denotes the prefactors in front of the integrals in (2.23), and $T_{\ell_i}^{ij\leftrightarrow kl}(\omega, z)$ equals to $T_{\ell_i}(s,t)$ with the parameterizations given in Table 1.

## 2.2  Implications of UV/IR behavior of $T(\omega, z)$

As stressed in [21], the analytic property of the celestial amplitude in complex $\beta$ plane reflects the UV and IR properties of $T(\omega, z)$. Since we will be performing a Mellin transform with respect to the center of mass energy $\omega$, special attention will be paid to the region $\omega \to 0$ and $\infty$ for which the integral might diverge. In limit $\omega \to 0$, we probe the IR limit of $T(\omega, z)$. Suppressing the massless logs for now, the amplitude takes the form

$$T(\omega, z)|_{\omega\to 0} = T_{\text{massless}}(\omega, z) + \sum_{p=0}^\infty g_p(z)\omega^{2p}\,. \qquad (2.25)$$

In $s$-kinematics, where $1 \geq z$, $g_p(z)$ are polynomial functions of at most degree $2p$ in $\frac{1-z}{z}$ reflecting the presence of contact interactions, i.e. higher dimension operators in the EFT description. The function $T_{\text{massless}}(\omega, z)$ summarizes the contribution from the massless particle exchange, which contain poles in $\frac{1-z}{z}$ and is of degree $\omega^0$ or $\omega^2$ for photon and graviton exchange respectively. Thus we see that the low energy amplitude is essentially a polynomial expansion in $\omega^2$. These generate single poles in $\beta$ since

$$\Psi(\beta, z) \sim \int_0^\Lambda d\omega\, \omega^{\beta-1}(\omega^{2p}) + \cdots = \frac{\Lambda^{\beta+2p}}{\beta+2p} + \cdots\,, \qquad (2.26)$$

where we only consider the part of the integral where $\omega \in [0, \Lambda]$. Thus we see that $\Psi(\beta, z)$ will have simple poles at $\beta = 0, -2, -4, \cdots$. When the massless loops are involved, the simple poles are then promoted to higher degree as discused in [21].

We now turn to the opposite limit, where $\omega \to \infty$ corresponding to fixed angle hard scattering,

$$s \to +\infty \quad \text{with} \quad \frac{s}{t} = \frac{z}{1-z} \quad \text{fixed.} \qquad (2.27)$$

We will exam two asymptotic behaviors of $T(s,t)$ in this limit,

$$T(\omega, z) \sim \begin{cases} g_p(z)\omega^{2p} + g_{p-1}(z)\omega^{2p-2} + \cdots & \text{power law,} \\ e^{-\omega^2 f(z)} & \text{exponential decay.} \end{cases} \qquad (2.28)$$

As discussed in [21], the second scenario is expected for amplitudes in gravitational theories due to black hole production. Indeed this is the case for string theory, which we will discuss in detail later on. In such case $\Psi(\beta, z)$ is convergent for $\text{Re}[\beta] > 0$. For the power law, following similar analysis as the IR region we again arrive at poles for

$\beta = -2p, -2p+2, -2p+4, \cdots$. Now we would like to have a celestial amplitude that is meromorphic in $\beta$, thus having a region of convergence. From the previous IR analysis, we see that $\Psi(\beta, z)$ will have poles at $\mathrm{Re}[\beta] \leq 0$, this suggest that we must have $p < 0$ such that there is a convergent region $0 < \mathrm{Re}[\beta] < -p$.[2]

## 2.3 Example: The massive scalar exchange

As a simple example for the above analysis, consider a massless scalar $\phi$ coupled to a massive scalar $X$ via a cubic coupling $g\phi^2 X$. The tree level 4-point scattering amplitude is

$$T(s,t) = -g^2 \left( \frac{1}{s - m^2 + i\varepsilon} + \frac{1}{u - m^2 + i\varepsilon} + \frac{1}{t - m^2 + i\varepsilon} \right). \tag{2.30}$$

The Mellin integral (2.24) converges when $\beta$ is bounded by

$$0 \leq \beta \leq 2. \tag{2.31}$$

Indeed for the $12 \leftrightarrow 34$ kinematics, by rewriting $\Psi^{12\leftrightarrow34}(\beta, z)$ as integrating from $\omega = -\infty$ to $\omega = \infty$,

$$\Psi^{12\leftrightarrow34}(\beta, z) = \frac{2^{3-\beta} z^2}{(1 - e^{i\pi\beta})} \int_{-\infty}^{\infty} d\omega\, \omega^{\beta-1} T\left( \omega^2, -\frac{(z-1)}{z}\omega^2 \right) \tag{2.32}$$

we can close the contour upward and pick up the residues giving,

$$\Psi_{\mathrm{scalar}}^{12\leftrightarrow34}(\beta, z) = \frac{\pi g^2}{\sin \frac{\pi\beta}{2}} \left( \frac{m}{2} \right)^{\beta-2} z^2 \left[ e^{\frac{1}{2}\pi i\beta} + z^{\frac{\beta}{2}} + \left( \frac{z}{z-1} \right)^{\frac{\beta}{2}} \right] \quad (z \geq 1). \tag{2.33}$$

Note that indeed starting from $\beta = 2$, one has simple poles at $\beta = 2, 4, 6, \cdots$, reflecting the divergence in UV. Similarly, below $\beta = 0$, one has simple poles at $\beta = 0, -4, -6, \cdots$, reflecting the EFT coefficients. The absence of $\beta = -2$ is due to the corresponding EFT operator vanishes on-shell, $s+t+u = 0$.

Importantly, there is a non-trivial imaginary part,[3]

$$\mathbf{Im}\, \Psi_{\mathrm{scalar}}^{12\leftrightarrow34}(\beta, z) = \pi g^2 \left( \frac{m}{2} \right)^{\beta-2} z^2 = 2^{3-\beta} \pi z^2 g^2 \mathbf{Res}_{\omega=m} \left[ \omega^{\beta-1} T(\omega, z) \right], \tag{2.34}$$

which is non-zero for any value of $\beta$. Furthermore for $s$-channel kinematics, only the physical threshold in the $s$-channel propagator contributes to imaginary part, while the $t, u$ diagram only contribute to the real part. Thus we see that the imaginary part of the celestial amplitude encodes the information of bulk factorization. This will be the focus of the next section.

---

[2]Indeed for $p = 0$ such as $\lambda\phi^4$ theory, the function $\Psi(\beta, z)$ cease to be meromorphic:

$$\int_0^{\infty} \frac{d\omega}{\omega} \omega^{i\epsilon} = \int_{-\infty}^{\infty} dx\, e^{i\epsilon x} = \delta(i\epsilon). \tag{2.29}$$

[3]Here and throughout this paper, we assume that $\beta$ is real when taking the imaginary part of $\Psi(\beta, z)$.

## 3 The imaginary part of the celestial amplitude

Here we would like to pose the following question: given a celestial amplitude $\Psi(\beta, z)$, what are the properties that reflect its origin as a local flat-space scattering amplitude. Already in the massive scalar case we've seen that for the $s$-channel kinematics, the fact that a massive scalar was being exchanged is reflected in the imaginary part of the celestial amplitude. In this section we will systematically study this property.

### 3.1 Bulk-locality to the imaginary part of $\Psi(\beta, z)$

The imaginary part of the amplitude is deeply rooted in causality, where the time ordered two point function introduces the $i\varepsilon$ prescription for the Feynman propagator. Indeed this is the origin of the imaginary piece of the scalar exchange $\Psi^{12\leftrightarrow34}(\beta, z)$, appearing in the $s$-channel where the internal particle can be interpreted as on-shell and moving forward in time. Thus to capture the imaginary piece, it will be useful to consider the Mellin transform as a contour integral on the complex $\omega$-plane.

Let us focus on the $12 \leftrightarrow 34$ kinematics. From previous discussions, we have seen that for $\Psi^{12\leftrightarrow34}(\beta, z)$ to be a meromorphic function, $T^{12\leftrightarrow34}(\omega, z)$ must vanish as $\omega \to \pm\infty$, i.e. $T^{12\leftrightarrow34}(\omega, z)$ behavior asymptotically as (2.28) for $p < 0$. In general, such an asymptotic behavior does not hold when $\omega$ approaches complex infinity, i.e. $\omega \to \infty \times e^{i\theta}$ for $\theta \neq 0, \pi$.

We will assume that the asymptotic behaviors in (2.28), which is defined for real $\omega$, can be extended for a small range of $\arg \omega$,

$$\arg \omega \in \left( -\theta_c^{12\leftrightarrow34}, \theta_c^{12\leftrightarrow34} \right) \cup \left( \pi - \theta_c^{12\leftrightarrow34}, \pi + \theta_c^{12\leftrightarrow34} \right), \tag{3.1}$$

with a finite angle $\theta_c^{12\leftrightarrow34}$ that in general depends on $z$. As we will see, in the case of open and closed string amplitudes, $\theta_c$s are in general finite and non-vanishing. Let us exam the analytic structure of the amplitude $T^{12\leftrightarrow34}(\omega, z)$ on the complex $\omega$-plane. By the bulk-locality, the exchange of single-particle states (of masses $m_i$) leads to $s$-channel poles located slight below or above the real axes (at $\omega = \pm m_i \pm i\epsilon$) due to Feynman $i\epsilon$. At the loop level, the exchange of multi-particle states lead to branch cuts located slight below or above the real axes. The crossing images of the $s$-channel poles are the $t$- and $u$-channel poles (at $\omega = \pm i\sqrt{\frac{z}{z-1}} m_i$ and $\omega = \pm i\sqrt{z} m_i$). The poles and branch cuts are shown schematically in figure 1. There could be other branch cuts away from the real and the imaginary axes, which do not correspond to the exchange of multi-particle states.[4] They are not depicted in the figures, as they would not play a role in our later computation. Finally, the branch cut of the $\omega^{\beta-1}$ is chosen to be along the negative real axis, and is also not depicted in the figures.

Consider the fan-like contour displayed in the left of figure 1, with an angle less than $\theta_c$. As the infinity part $I_3$ vanishes, the absence of poles in the contour imply

$$\Psi(\beta, z) = B(z) \int_{I_1} d\omega \, \omega^{\beta-1} T(\omega, z) = -B(z) \int_{I_2} d\omega \, \omega^{\beta-1} T(\omega, z). \tag{3.2}$$

---

[4]The branch points of such branch cuts are called the anomalous thresholds.

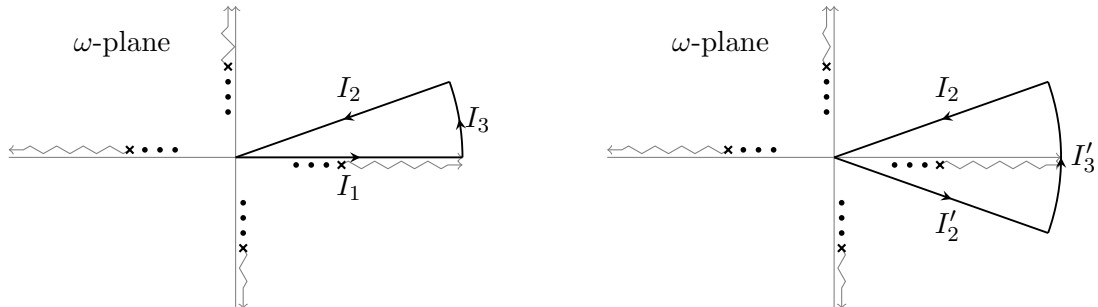

**Figure 1**. The analytic structure of the amplitude $T(\omega, z)$ on the complex $\omega$ plane. The branch cuts associated to the anomalous thresholds are not depicted in the figure, since they are away from the real $\omega$ axes, and would not contribute the the contour integrals when the angle between the $I_2$ and $-I_2'$ segments of the contour is small enough.

Now extend the contour symmetrically to the lower half plane, as shown in the right of figure 1. The new contour will then pick up the residue from the $i\varepsilon$ prescription of the propagators. Note that since $I_2'$ is just the reflection of $I_2$ along the real axes, the two simply have the opposite sign for the real part:

$$\int_{I_2'} d\omega\, \omega^{\beta-1} T(\omega, z) = -\Big[\int_{I_2} d\omega\, \omega^{\beta-1} T(\omega, z)\Big]^*. \tag{3.3}$$

Furthermore, since the $i\varepsilon$ poles are away from the contour $I_2'$, and we are free to take the $\varepsilon \to 0$ limit, for which the amplitude $T(\omega, z)$ is a real function of $\omega$, i.e.

$$T(\omega, z)^* = T(\omega^*, z). \tag{3.4}$$

Again since the integration along $I_3'$ vanishes, the imaginary part of $\Psi(\beta, z)$ is now simply given by the poles and branch cuts from the $i\varepsilon$ prescription

$$\begin{aligned}
\mathbf{Im}\, \Psi(\beta, z) &= -\frac{1}{2i} B(z) \int_{I_2+I_2'} d\omega\, \omega^{\beta-1} T(\omega, z) \\
&= -B(z) \left\{ \pi \sum_i \operatorname*{\mathbf{Res}}_{\omega \to m_i} \Big[\omega^{\beta-1} T(\omega, z)\Big] + \int_M^\infty d\omega\, \omega^{\beta-1} \mathbf{Disc}\, [T(\omega, z)] \right\},
\end{aligned} \tag{3.5}$$

where $m_i$ are the position of the factorization poles, and $M$ is the branch point of the branch cuts, which is present due to loop effects.[5]

In summary, we find that the imaginary part of the celestial amplitude is simply governed by the residue or (massive) discontinuity of the scattering amplitude. For $s$-channel kinematics, we find that $\mathbf{Im}[\Psi]$ is given by the projection of the $s$-channel residue or discontinuity onto the celestial sphere as in eq.(3.5).

---

[5]If the convergent angle $\theta_0$ is great than $\frac{\pi}{2}$ (e.g. massive scalar exchange amplitude), one might worry that there will be poles coming from other channel contributing to the imaginary part in (3.5). In fact, because other channel poles are lying on the pure imaginary axis, and they are coming in pair. One can easily verify total residue sum is 0 in the RHS contour of fig. 1. Thus the result is not changed even if the $\theta_0 > \pi/2$ is bigger than $\pi/2$

## 3.2 Positivity in $\mathbf{Im}[\Psi(\beta, z)]$

If the external states are arranged as $a, b \to b, a$, where $(a, b)$ represents potentially distinct species, the residue and the discontinuity are positively expandable on Legendre polynomials for scalars and Jacobi polynomials for spinning states [23]. Thus the imaginary part of the celestial amplitude must be positively expanded on the Mellin-transform of these orthogonal polynomials.

At the tree-level, the imaginary part of the celestial amplitude only picks up the poles corresponding to the factorization channels. The associated residues in the three kinematic regimes are

$$12 \to 34: \quad \mathbf{Res}_{\omega=m}\left[\omega^{\beta-1}\frac{m^{2J}P_J(\frac{u-t}{m^2})}{\omega^2 - m^2}\right] = \frac{m^{\beta-2+2J}}{2} \cdot P_J\left(\frac{z-2}{z}\right),$$

$$13 \to 24: \quad \mathbf{Res}_{\omega=m}\left[\omega^{\beta-1}\frac{m^{2J}P_J(\frac{s-t}{m^2})}{\omega^2 - m^2}\right] = \frac{m^{\beta-2+2J}}{2} \cdot P_J(1 - 2z), \tag{3.6}$$

$$14 \to 23: \quad \mathbf{Res}_{\omega=m}\left[\omega^{\beta-1}\frac{m^{2J}P_J(\frac{u-s}{m^2})}{\omega^2 - m^2}\right] = \frac{m^{\beta-2+2J}}{2} \cdot P_J\left(\frac{z+1}{z-1}\right).$$

Note that we have written the argument of the Legendre polynomials in terms or Mandelstam invariants in such a way that manifest the exchange symmetry in each channel. Thus from (3.5) the imaginary part of the scalar celestial amplitude must be positively expanded on the following basis

$$\mathbf{Im}\,\Psi^{12\leftrightarrow34}(\beta, z) = \pi z^2 \sum_{i\in\mathcal{I}} p_i P_{J_i}\left(\frac{z-2}{z}\right) \qquad\qquad (z \geq 1),$$

$$\mathbf{Im}\,\Psi^{13\leftrightarrow24}(\beta, z) = \pi z^{\frac{\beta}{2}+2} \sum_{i\in\mathcal{I}} p_i P_{J_i}(1 - 2z) \qquad\qquad (1 \geq z \geq 0), \tag{3.7}$$

$$\mathbf{Im}\,\Psi^{14\leftrightarrow23}(\beta, z) = \pi(-z)^{\frac{\beta}{2}+2}(1 - z)^{-\frac{\beta}{2}} \sum_{i\in\mathcal{I}} p_i P_{J_i}\left(\frac{z+1}{z-1}\right) \quad (0 \geq z),$$

where $p_i = g_i^2 \left(\frac{m_i}{2}\right)^{\beta-3} m_i^{2J_i+1}$.

Extension to spinning external states is straight forward. On the factorization pole, the residue polynomial is now:

$$T_{\ell_i}(s, t)|_{s\to m^2} = -\frac{m^{2J}d^J_{\ell_{34}, \ell_{12}}\left(\arccos(\frac{u-t}{m^2})\right)}{s - m^2 + i\varepsilon}, \tag{3.8}$$

where $\ell_{ij\pm kl} = (\ell_i - \ell_j) \pm (\ell_k - \ell_l)$, and $d^J_{\ell_{34}, \ell_{12}}(\phi)$ is the Wigner (small) $d$-matrix which can be conveniently given in Jacobi polynomials,

$$d^J_{\ell_{34}, \ell_{12}}(\phi) = B^J_{12;34}\left(\sin\frac{\phi}{2}\right)^{\ell_{12-34}}\left(\cos\frac{\phi}{2}\right)^{\ell_{12+34}} \mathcal{J}^{\ell_{12-34}, \ell_{12+34}}_{J-\ell_{12}}(\cos\phi), \tag{3.9}$$

where $\mathcal{J}^{\alpha,\beta}_n$ is the Jacobi polynomial and the constant $B^J_{i_1i_2;i_3i_4} \equiv \sqrt{\frac{(J+\ell_{i_1i_2})!(J-\ell_{i_1i_2})!}{(J_i+\ell_{i_3i_4})!(J-\ell_{i_3i_4})!}}$. This gives the following representation of the imaginary part $\mathbf{Im}\left[\Psi(\beta, z)\right]$ in the three different

regions (2.21) on the equator:

$$\mathbf{Im}\,\Psi^{12\leftrightarrow34}(\beta,z) = \sum_i p_{i12;34}\;\; z^{2-\ell_{12}}(z-1)^{\frac{\ell_{12-34}}{2}}\,\mathcal{J}^{\ell_{12+34},\ell_{12-34}}_{J_i-\ell_{12}}\left(\frac{z-2}{z}\right) \qquad (z\geq 1)\,,$$

$$\mathbf{Im}\,\Psi^{13\leftrightarrow24}(\beta,z) = \sum_i p_{i13;24}\;\; z^{\frac{\beta+\ell_{13+24}}{2}+2}(1-z)^{\frac{\ell_{13-24}}{2}}\,\mathcal{J}^{\ell_{13+24},\ell_{13-24}}_{J_i-\ell_{13}}(1-2z) \quad (1\geq z\geq 0)\,,$$

$$\mathbf{Im}\,\Psi^{14\leftrightarrow23}(\beta,z) = \sum_i p_{i14;23}\;\; \frac{(-z)^{\frac{\beta+\ell_{14+23}}{2}+2}}{(1-z)^{\frac{\beta}{2}+\ell_{23}}}\,\mathcal{J}^{\ell_{14+23},\ell_{14-23}}_{J_i-\ell_{23}}\left(\frac{z+1}{z-1}\right) \qquad (0\geq z)\,,$$

$$\text{(3.10)}$$

where $p_{i12;34} = \pi g_i\left(\frac{m}{2}\right)^{\beta-3}m_i^{2J_i+1}B^{J_i}_{12;34} > 0$. Indeed setting $\ell_i = 0$ in (3.10) one recovers the scalar basis in (3.7). The coefficient $g_i$ becomes positive if the external states are arranged as $\ell_3 = -\ell_2$ and $\ell_4 = -\ell_1$, i.e. such that the configuration corresponds to forward scattering.

Thus for each kinematics, $\mathbf{Im}\,\Psi$ has a positive expansion on the basis in eq.(3.10). Note that the fact that the Wigner $d$-matrix serves as an expansion basis for the flat-space amplitude is a reflection of factorization and Poincaré symmetry. Since the basis functions in eq.(3.10) is simply the projection of the Wigner $d$-matrix, they must be intimately tied to the "image" of Poincaré symmetry on the celestial sphere. Indeed, in the following, we will find that these are precisely the Poincaré partial waves introduced in [20].

## 3.3 Poincaré partial wave expansion

Let us first review the Poincaré partial waves introduced in [20], focusing on the celestial amplitude in the $12 \leftrightarrow 34$ kinematics. It can be written as a inner product between the in and out states

$$\tilde{\mathcal{A}}^{12\leftrightarrow34}_{\Delta_i,\ell_i}(z_i,\bar{z}_i) = \langle\Delta_3,z_3,\bar{z}_3,\ell_3,\Delta_4,z_4,\bar{z}_4,\ell_4|\Delta_1,z_1,\bar{z}_1,\ell_1,\Delta_2,z_2,\bar{z}_2,\ell_2\rangle\,. \qquad (3.11)$$

The Hilbert space can be decomposed into irreducible unitary representations of the Poincaré algebra. According to Wigner's classification, they are the massive representations labeled by the mass $m$ and spin $J$, and the massless representations labeled by the helicity $\ell$. Consider the projectors

$$\mathbb{P}_{m,J} = \frac{1}{2J+1}\sum_{J_3=-J}^{J}|m,J,J_3\rangle\langle m,J,J_3|\,,$$

$$\mathbb{P}_\ell = \frac{1}{2}\sum_{\epsilon=\pm}|\epsilon\ell\rangle\langle\epsilon\ell|\,, \qquad (3.12)$$

which project onto a single massive or massless representation. The projectors $\mathbb{P}_{m,J}$ and $\mathbb{P}_\ell$ commute with the Poincaré generators $P^\mu$ and $M^{\mu\nu}$, i.e.

$$[P^\mu,\mathbb{P}_{m,J}] = 0 = [M^{\mu\nu},\mathbb{P}_{m,J}]\,, \qquad (3.13)$$

and similarly for the massless projector $\mathbb{P}_\ell$. The massive and massless Poincaré partial waves are defined by inserting the projectors into the inner product (3.11),

$$
\begin{aligned}
\tilde{\mathcal{A}}^{12\leftrightarrow34}_{\Delta_i,\ell_i;m,J}(z_i,\bar{z}_i) &= \langle \Delta_3,z_3,\bar{z}_3,\ell_3,\Delta_4,z_4,\bar{z}_4,\ell_4|\mathbb{P}_{m,J}|\Delta_1,z_1,\bar{z}_1,\ell_1,\Delta_2,z_2,\bar{z}_1,\ell_2\rangle\,, \\
\tilde{\mathcal{A}}^{12\leftrightarrow34}_{\Delta_i,\ell_i;\ell}(z_i,\bar{z}_i) &= \langle \Delta_3,z_3,\bar{z}_3,\ell_3,\Delta_4,z_4,\bar{z}_4,\ell_4|\mathbb{P}_\ell|\Delta_1,z_1,\bar{z}_1,\ell_1,\Delta_2,z_2,\bar{z}_1,\ell_2\rangle\,.
\end{aligned}
\tag{3.14}
$$

The translation and Lorentz generators $P^\mu$ and $M^{\mu\nu}$ act on the massless single particle states as differential operators $\mathcal{P}^\mu$ and $\mathcal{M}^{\mu\nu}$ given explicitly in [12], which we list in Appendix B. By (3.13), the Poincaré partial waves satisfy the constraints

$$
\begin{aligned}
(\mathcal{P}_1^\mu + \mathcal{P}_2^\mu - \mathcal{P}_3^\mu - \mathcal{P}_4^\mu)\tilde{\mathcal{A}}^{12\leftrightarrow34}_{\Delta_i,\ell_i;m,J}(z_i,\bar{z}_i) &= 0\,, \\
(\mathcal{M}_1^{\mu\nu} + \mathcal{M}_2^{\mu\nu} + \mathcal{M}_3^{\mu\nu} + \mathcal{M}_4^{\mu\nu})\tilde{\mathcal{A}}^{12\leftrightarrow34}_{\Delta_i,\ell_i;m,J}(z_i,\bar{z}_i) &= 0\,,
\end{aligned}
\tag{3.15}
$$

and similar for the massless Poincaré partial waves. Since the above constraints are the same constraints used in [17] to derive the form in (2.12) and (2.14) of the celestial amplitude, the Poincaré partial waves should also take the form as (2.12) and (2.14).

Beside (3.15), the Poincaré partial waves satisfy additional differential equations given by the Casimir operators of the Poincaré algebra. The Poincaré algebra has a quadratic and a quartic Casimir operators

$$
P_\mu P^\mu\,, \quad W_\mu W^\mu\,,
\tag{3.16}
$$

where $W^\mu = \frac{1}{2}\epsilon^{\mu\nu\rho\sigma}M_{\nu\rho}P_\sigma$ is the Pauli-Lubanski pseudo-vector. The two Casimir operators have eigenvalues $-m^2$ and $m^2 J(J+1)$ when acting on a state with mass $m$ and spin $J$, i.e.

$$
\begin{aligned}
P_\mu P^\mu |m,J,J_3\rangle &= -m^2 |m,J,J_3\rangle\,, \\
W_\mu W^\mu |m,J,J_3\rangle &= m^2 J(J+1) |m,J,J_3\rangle\,.
\end{aligned}
\tag{3.17}
$$

By inserting the Casimir operator $P^\mu P_\mu$ into the inner product form of the Poincaré partial wave $\tilde{\mathcal{A}}^{12\leftrightarrow34}_{\Delta_i,\ell_i;m,J}(z_i,\bar{z}_i)$, and using the formulae (B.1) and (B.3), one find a differential equation

$$
\begin{aligned}
(\mathcal{P}_1 &+ \mathcal{P}_2)^\mu(\mathcal{P}_1 + \mathcal{P}_2)_\mu\tilde{\mathcal{A}}^{12\leftrightarrow34}_{\Delta_i,\ell_i;m,J}(z_i,\bar{z}_i) \\
&= \langle \Delta_3,z_3,\bar{z}_3,\ell_3,\Delta_4,z_4,\bar{z}_4,\ell_4|\mathbb{P}_{m,J}P^\mu P_\mu|\Delta_1,z_1,\bar{z}_1,\ell_1,\Delta_2,z_2,\bar{z}_1,\ell_2\rangle \\
&= -m^2 \tilde{\mathcal{A}}^{12\leftrightarrow34}_{\Delta_i,\ell_i;m,J}(z_i,\bar{z}_i)\,.
\end{aligned}
\tag{3.18}
$$

Similarly, the Casimir operator $W^\mu W_\mu$ leads to another differential equation

$$
(\mathcal{W}_1 + \mathcal{W}_2)^\mu(\mathcal{W}_1 + \mathcal{W}_2)_\mu\tilde{\mathcal{A}}^{12\leftrightarrow34}_{\Delta_i,\ell_i;m,J}(z_i,\bar{z}_i) = m^2 J(J+1)\tilde{\mathcal{A}}^{12\leftrightarrow34}_{\Delta_i,\ell_i;m,J}(z_i,\bar{z}_i)\,,
\tag{3.19}
$$

where $\mathcal{W}^\mu = \frac{1}{2}\epsilon^{\mu\nu\rho\sigma}\mathcal{M}_{\nu\rho}\mathcal{P}_\sigma$. Factoring out some conformal factors similar to (2.12) and (2.14) as

$$
\begin{aligned}
\tilde{\mathcal{A}}^{12\leftrightarrow34}_{\Delta_i,\ell_i;m,J}(z_i,\bar{z}_i) = &\frac{\left(\frac{z_{14}}{z_{13}}\right)^{h_3-h_4}\left(\frac{z_{24}}{z_{14}}\right)^{h_1-h_2}\left(\frac{\bar{z}_{14}}{\bar{z}_{13}}\right)^{\bar{h}_3-\bar{h}_4}\left(\frac{\bar{z}_{24}}{\bar{z}_{14}}\right)^{\bar{h}_1-\bar{h}_2}}{z_{12}^{h_1+h_2}z_{34}^{h_3+h_4}\bar{z}_{12}^{\bar{h}_1+\bar{h}_2}\bar{z}_{34}^{\bar{h}_3+\bar{h}_4}} \\
&\times (z-1)^{\frac{\Delta_1-\Delta_2-\Delta_3+\Delta_4}{2}}\delta(iz-i\bar{z})\Phi^{12\leftrightarrow34}_{m,J}(\boldsymbol{\Delta},\ell_i,z)\,,
\end{aligned}
\tag{3.20}
$$

(3.18) reduces to the differential equation on the total dimension $\boldsymbol{\Delta}$ and the cross ratio $z$

$$-4e^{2\partial\boldsymbol{\Delta}}\Phi_{m,J}^{12\leftrightarrow34}(\boldsymbol{\Delta},\ell_i,z) = -m^2\Phi_{m,J}^{12\leftrightarrow34}(\boldsymbol{\Delta},\ell_i,z)\,. \tag{3.21}$$

Similarly, (3.19) gives the differential equation

$$\left[\frac{(\frac{1}{4}\ell_{12+34}^2-4)z^2+(10-\ell_{12}\ell_{34})z-6}{z-1} + (3z-4)z\partial - (z-1)z^2\partial^2\right]\Phi_{m,J}^{12\leftrightarrow34}(\boldsymbol{\Delta},\ell_i,z)$$
$$= J(J+1)\Phi_{m,J}^{12\leftrightarrow34}(\boldsymbol{\Delta},\ell_i,z)\,. \tag{3.22}$$

Let us briefly comment on the Casimir equations for the massless Poincaré partial waves. By the same derivation, we find the Casimir operator $P^\mu P_\mu$ gives the differential equation (3.21) with $m^2 = 0$. However, the differential operator on the right hand side of (3.21) is invertible. Hence, the massless Poincaré partial waves should be zero identically.

One could repeat the above analysis for the celestial amplitude in the $13 \leftrightarrow 24$ and $14 \leftrightarrow 23$ kinematics, and derive the corresponding differential equations for the Poincaré partial waves $\Phi_{m,J}^{13\leftrightarrow24}(\boldsymbol{\Delta},\ell_i,z)$ and $\Phi_{m,J}^{14\leftrightarrow23}(\boldsymbol{\Delta},\ell_i,z)$. The solutions to the Casimir equations in the three kinematics (2.21) are (again $\beta = \boldsymbol{\Delta} - 4$)

$$\Phi_{m,J}^{\ell_i}(\beta,z) = \left(\frac{m}{2}\right)^\beta\sqrt{\frac{(2J+1)}{m}}\begin{cases} B_{12,34}^J z^{2-\ell_{12}}(z-1)^{\frac{\ell_{12-34}}{2}}\mathcal{J}_{J-\ell_{12}}^{\ell_{12+34},\ell_{12-34}}\left(\frac{z-2}{z}\right) & (z \geq 1)\,, \\ B_{13,24}^J z^{\frac{\beta+\ell_{13+24}}{2}+2}(1-z)^{\frac{\ell_{13-24}}{2}}\mathcal{J}_{J-\ell_{13}}^{\ell_{13+24},\ell_{13-24}}(1-2z) & (1 \geq z \geq 0)\,, \\ B_{14,23}^J \frac{(-z)^{\frac{\beta+\ell_{14+23}}{2}+2}}{(1-z)^{\frac{\beta}{2}+\ell_{23}}}\mathcal{J}_{J-\ell_{23}}^{\ell_{14+23},\ell_{14-23}}\left(\frac{z+1}{z-1}\right) & (0 \geq z)\,. \end{cases} \tag{3.23}$$

One see exactly that the Poincaré partial waves (3.23) are identical to the expansion basis in (3.10) in each kinematics threshold up to normalization factors.

Thus combining everything, we conclude that the imaginary part of the celestial amplitude has a positive expansion on the massive Poincaré partial wave, namely,

$$\mathbf{Im}\,\Psi(\beta,\ell_i,z) = \sum_a p_a\Phi_{m_a,J_a}(\beta,\ell_i,z)\,, \quad p_a > 0\,. \tag{3.24}$$

Note that since the Jacobi polynomials are orthogonal polynomials, the Poincaré partial waves also satisfy orthogonality relations [20], and the coefficients $p_a$ are unique. The generalization to supersymmetric theories is straightforward. Once again the expansion polynomial is given by the projection of the susyspinning polynomials on to the celestial sphere. These polynomials are again Jacobi polynomials, but with the helicities indicating that of the highest weighted state in a susy multiplet. For details see [24].

**Crossing symmetry:** Let us consider the implications of crossing symmetry on the celestial amplitude. For a crossing symmetric amplitude, we would expect the three kinematic region of the celestial amplitude to be related to each other. Indeed, we have:

$$\begin{cases} s \leftrightarrow u & \Psi^{13\leftrightarrow24}(\beta,z) = z^{\frac{\beta}{2}+4}\Psi^{12\leftrightarrow34}(\beta,\frac{1}{z}) & (1 \geq z \geq 0)\,, \\ s \leftrightarrow t & \Psi^{12\leftrightarrow34}(\beta,z) = \left(\frac{z}{1-z}\right)^{\frac{\beta}{2}+2}\Psi^{14\leftrightarrow23}(\beta,1-z) & (z \geq 1)\,, \\ u \leftrightarrow t & \Psi^{13\leftrightarrow24}(\beta,z) = (z-1)^2\Psi^{14\leftrightarrow23}(\beta,\frac{z}{z-1}) & (1 \geq z \geq 0)\,. \end{cases} \tag{3.25}$$

for the celestial amplitude of any permutation invariant theory. Furthermore, since the residue of each channel must respect the exchange symmetry of the legs on one side of the factorization, this implies that the imaginary part of $\Psi(\beta, z)$ must further satisfy the following "self-dual" like identities:

$$
\begin{cases}
\mathbf{Im}\,\Psi^{13\leftrightarrow24}(\beta, z) = \left(\frac{z}{1-z}\right)^2 \mathbf{Im}\,\Psi^{13\leftrightarrow24}(\beta, 1-z) & (1 \geq z \geq 0)\,, \\
\mathbf{Im}\,\Psi^{14\leftrightarrow23}(\beta, z) = (-1)^{\frac{\beta}{2}} z^{\frac{3\beta}{2}+4} \mathbf{Im}\,\Psi^{14\leftrightarrow23}(\beta, \frac{1}{z}) & (0 \geq z)\,, \\
\mathbf{Im}\,\Psi^{12\leftrightarrow34}(\beta, z) = (z-1)^2 \mathbf{Im}\,\Psi^{12\leftrightarrow34}\left(\beta, \frac{z}{z-1}\right) & (z \geq 1)\,.
\end{cases}
\tag{3.26}
$$

It is straightforward to verify that the Poincaré partial waves in (3.23) respect the crossing equations (3.25) and (3.26).[6]

## 4  Open and closed string celestial amplitudes

We have already seen how the imaginary part of the celestial amplitude encodes the information of poles and branch cuts on complex $\omega$ plane. Furthermore, as discussed in Section 5, the integrand can be analytic continued to other unphysical $z$ domains. However, the UV convergence of the integral relies on certain regions of $\omega$ plane. In this section, we will first use type-I and II superstring amplitudes:[7]

$$
\text{type-I}: \quad \frac{\Gamma[-s]\Gamma[-t]}{\Gamma[1+u]}, \qquad \text{type-II}: \quad \frac{\Gamma[-s]\Gamma[-t]\Gamma[-u]}{\Gamma[1+s]\Gamma[1+u]\Gamma[1+t]}\,,
\tag{4.1}
$$

as examples to determine the convergent regions on $\omega$ plane for all $z$ dependance. As indicated in Figure 2, they are in general $z$-dependent for open strings while enjoys $z$-independent for closed strings. Then we will discuss how bulk-locality encoded in the imaginary part of tree-level celestial string amplitude in physical $z$ domains. Finally, when we discuss the celestial dispersion relation in Section 6 we will verify those relations with celestial string amplitudes in Section 6.1.

### 4.1  Convergent regions in string theory

As is widely known, in the fixed-angle scatterings, string amplitudes are expected to decay exponentially at large real $\omega$. However, the suppression is in general not true in the whole complex $\omega$ plane. Therefore in this sub-section we will study this problem for open and closed string amplitudes. Writing $\omega = re^{i\theta}$, the $r \to \infty$ limit of string amplitudes behave as:

$$
T(r^2 e^{2i\theta}, z) \sim \exp\left[g(\theta, z)r^2 + \mathcal{O}(\log r)\right].
\tag{4.2}
$$

Hence, for given $z$, the amplitude is suppressed in the limit $r \to \infty$ when

$$
\mathbf{Re}\left[g(\theta, z)\right] < 0\,,
\tag{4.3}
$$

---

[6]Note that eq.(3.25) only holds for scalar blocks of even spin. For odd spins there will be an additional sign as expected.

[7]Here we have neglected the $F^4$ and $R^4$ pre-factor for simplicity. Their effect can be incorporated straightforwardly.

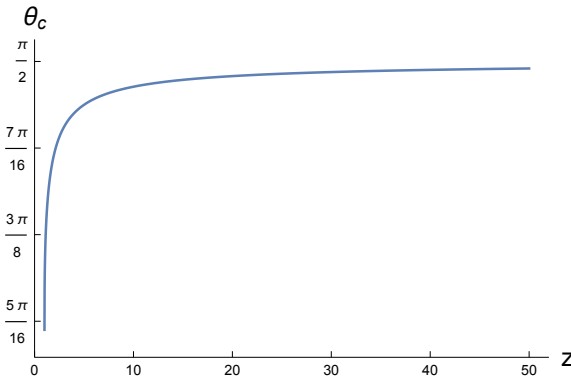

**Figure 2**. The angle $\theta_c$ for the open string amplitude in the $12 \leftrightarrow 34$ kinematics as a function of $z$.

which, as one will see, gives the convergent region

$$\theta \in (-\theta_c, \theta_c) \cup (\pi - \theta_c, \pi + \theta_c), \tag{4.4}$$

where the angle $\theta_c$ is the root of $\mathbf{Re}[g(\theta, z)] = 0$ when $\theta \in (0, \frac{\pi}{2})$. Let us consider the open and closed string separately,

- **Open string**

  The gluon amplitude in the type-I superstring theory is

  $$T_{\text{open}}(s, u) = \frac{\Gamma(-s)\Gamma(-t)}{\Gamma(1 - s - t)}. \tag{4.5}$$

  In the $12 \to 34$ kinematic region $(z > 1)$, and restricting to $\theta \in (0, \frac{\pi}{2})$, we have

  $$g(\theta, z) = e^{2i\theta} \frac{[(z-1)(\log(z - 1) + i\pi) - z\log(z)]}{z}. \tag{4.6}$$

  The inequality (4.3) gives the following equation for the angle $\theta_c$,

  $$\cot(2\theta_c) = \frac{\pi(1 - z)}{z\log z - (z - 1)\log(z - 1)}. \tag{4.7}$$

  The angle $\theta_c$ as a function of $z$ is plotted in figure 2. One can see that starting from $z = 1$, the angle $\theta_c$ is gradually increasing from $\frac{\pi}{4}$ to $\frac{\pi}{2}$ as $z \to \infty$.

  Also one can perform the same analysis for general angle $\theta \in (-\pi, \pi)$, and find that the convergent regions in the other quadrants are also specified by the same angle $\theta_c$ and given by (4.4). One can further repeat the same analysis for the other kinematics $13 \leftrightarrow 24$ and $14 \leftrightarrow 23$. The results are summarized in Table 2.

| $\theta_c^{12\leftrightarrow34}(z)$ $(z \geq 1)$ | $\theta_c^{13\leftrightarrow24}(z)$ $(1 \geq z \geq 0)$ | $\theta_c^{14\leftrightarrow23}(z)$ $(0 \geq z)$ |
|---|---|---|
| $\frac{1}{2}\cot^{-1}\left[\frac{\pi(1-z)}{z\log z-(z-1)\log(z-1)}\right]$ | $\frac{\pi}{4}$ | $\frac{1}{2}\cot^{-1}\left[\frac{\pi z}{(1-z)\log(1-z)+z\log(-z)}\right]$ |

**Table 2**. Characteristic angles in open string.

- **Closed string**

  The graviton amplitude in the type-II superstring is

  $$T_{\text{closed}}(s, u) = \frac{\Gamma(-s)\Gamma(-t)\Gamma(-u)}{\Gamma(1+s)\Gamma(1+t)\Gamma(1+u)} \tag{4.8}$$

  It turns out that the large $r$ behavior of the closed string amplitude is much simpler than the open string case. In the $12 \leftrightarrow 34$ kinematic region $(z \geq 1)$, we find

  $$g(\theta, z) = 2e^{2i\theta}\frac{(z-1)\log(z-1) - z\log z}{z}, \quad (0 < \theta < \pi/2). \tag{4.9}$$

  The condition for the real part to be negative is

  $$\cos(2\theta) > 0 \Rightarrow 0 < \theta < \frac{\pi}{4}. \tag{4.10}$$

  We can repeat the same analysis for general angle $\theta \in (-\pi, \pi)$ and for the other kinematics $13 \leftrightarrow 24$ and $14 \leftrightarrow 23$. The convergent regions for all cases are given by (4.4) with the angle $\theta_c$ equals to $\frac{\pi}{4}$.

## 4.2 $\text{Im}\,\Psi(\beta, z)$ in string theory

Taking $\beta \in \mathbb{R}$, from (3.5), $\text{Im}\,\Psi(\beta, z)$ is given as a sum of the residues of the poles (and also an integral of the discontinuity) near the positive real axis. For both open and closed string amplitudes, the poles near the positive real axis are located at $\omega = \sqrt{n}$ for $n \in \mathbb{Z}_{>0}$, and we have

$$\text{Im}\,\Psi(\beta, z) = -\pi B(z) \sum_{n=1}^{\infty} (\sqrt{n})^{\beta-1} \mathop{\text{Res}}_{\omega_n=\sqrt{n}} \Big[T(\omega_n, z)\Big] \quad (\beta > \beta_*) \tag{4.11}$$

In the following, we will verify this formula by directly comparing the truncated sum with the numerical result of the Mellin integration:

- **Open String**

  The massive poles of the open string amplitude (4.5) are located at $s, t = 1, 2, \ldots, n$. And the corresponding residues for $s$-channel are,

  $$\mathbf{Res}_{s=n}[T_{\text{open}}(s, t)] = \frac{\prod_{k=1}^{n-1}(t+k)}{n!}. \tag{4.12}$$

  The residues of the $t$-channel poles are given by exchanging $s$ with $t$. Now we can compute $\text{Im}\Psi(\beta, z)$. In $12 \to 34$ kinematic region, using eq.(3.5), the imaginary part will only receive contribution from $s$-channel massive poles located on the positive-$\omega$ axis at $\omega_n = \sqrt{n}$ with the residue

  $$\mathop{\mathbf{Res}}_{\omega=\sqrt{n}} \Big[T_{\text{open}}^{12\to34}\Big(\omega^2, -\frac{1}{z}\omega^2\Big)\Big] = \frac{\big[n\big(\frac{1}{z}-1\big)+1\big]_{n-1}}{2\sqrt{n}n!}. \tag{4.13}$$

Thus, the imaginary part of the open string celestial amplitude is

$$\mathbf{Im}\,\Psi_{\text{open}}^{12\leftrightarrow34}(\beta,z) = -\pi B^{12\to34}(z)\sum_{n=1}^{\infty}(\sqrt{n})^{\beta-1}\,\mathop{\mathbf{Res}}_{\omega=\sqrt{n}}\left[T_{\text{open}}^{12\to34}\left(\omega^2,-\frac{1}{z}\omega^2\right)\right]$$

$$= -\pi 2^{3-\beta}z^2\sum_{n=1}^{\infty}(\sqrt{n})^{\beta-2}\frac{\left[n\left(\frac{1}{z}-1\right)+1\right]_{n-1}}{2n!}\quad (z>1),\quad (4.14)$$

One can perform the same computation for $14\to23$, the result is,

$$\mathbf{Im}\,\Psi_{\text{open}}^{14\leftrightarrow23}(\beta,z) = -\pi B^{14\to23}(z)\sum_{n=1}^{\infty}(\sqrt{n})^{\beta-1}\,\mathop{\mathbf{Res}}_{\omega=\sqrt{n}}\left[T_{\text{open}}^{14\to23}\left(\frac{z}{1-z}\omega^2,-\frac{1}{z-1}\omega^2\right)\right]$$

$$= -\pi 2^{3-\beta}(-z)^{\frac{\beta}{2}+2}(1-z)^{-\frac{\beta}{2}}\sum_{n=1}^{\infty}(\sqrt{n})^{\beta-2}\frac{\left(\frac{nz}{1-z}+1\right)_{n-1}}{2n!}\quad (z<0).$$

$$(4.15)$$

Note that $\mathbf{Im}\,\Psi_{\text{open}}^{13\leftrightarrow24}(\beta,z)=0$ because the open string amplitude $T_{\text{open}}(s,t)$ has no $u$-channel pole. The comparison of the truncated sum of eq.(4.14) with the result from the numerical Mellin integral (2.24) is given in Figure 3. One can see that $\mathbf{Im}\,\Psi_{\text{open}}^{12\leftrightarrow34}(\beta,z)$ diverges in the limits $z\to\infty$ and $z\to1$. The divergent at $z\to\infty$ is due to the overall prefactor $B^{12\to34}(z)=\pi z^2$, while the divergent at $z\to1$ is due to the non-convergence of the residue sum reflecting the massless $1/t$ singularity in that limit. Note that the divergence occurs for positive $\beta$. For negative $\beta$ the imaginary part is finite as we will see in Section 6.2.

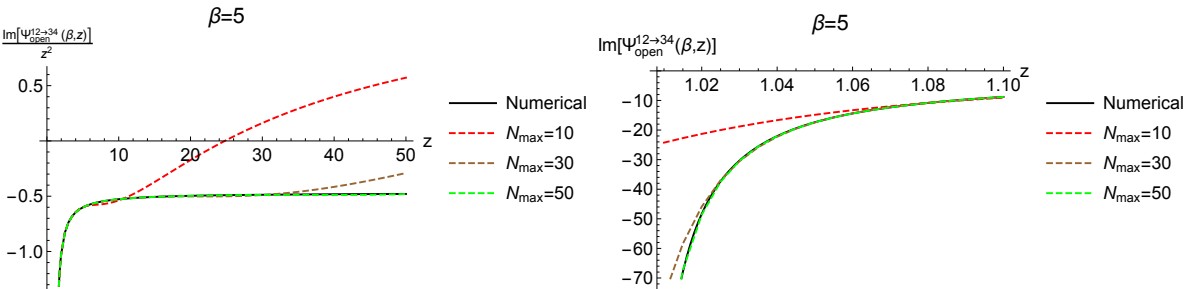

**Figure 3**. Here we present the comparison of numerical evaluation of Mellin integral and residue sum formula eq.(4.14) for the imaginary part of celestial open string amplitude. LHS and RHS shows the behavior of $\mathbf{Im}\,\Psi_{\text{open}}^{12\leftrightarrow24}(\beta,z)$ in $z\to\infty$ and $z\to1$

- **Closed String**

  In the closed string case, the residues at $s=1,2,\ldots n$ are

$$\mathbf{Res}_{s=n}[T_{\text{closed}}(s,t)] = \frac{\prod_{k=1}^{n-1}(t+k)^2}{(n!)^2}\quad (4.16)$$

Again, let us focus on the $12 \leftrightarrow 34$ kinematics and the region $z \geq 1$. Using eq.(3.5), the residue of the pole at $\omega = \sqrt{n}$ is

$$\operatorname*{Res}_{\omega=\sqrt{n}} \left[ T_{\text{closed}}(\omega^2, \frac{1-z}{z}\omega^2) \right] = \frac{\left[ \left( 1 + \frac{1-z}{z}n \right)_{n-1} \right]^2}{2\sqrt{n}n(n!)^2}. \tag{4.17}$$

Thus, the imaginary part of the massive contribution is given by

$$\mathbf{Im}\,\Psi_{\text{closed}}^{12\leftrightarrow34}(\beta, z) = -\pi 2^{3-\beta} z^2 \sum_{n=1}^{\infty} \frac{(\sqrt{n})^{\beta-2}}{2} \left( \frac{\left[ n\left(\frac{1}{z}-1\right)+1 \right]_{n-1}}{n!} \right)^2 \tag{4.18}$$

The comparison with result from the numerical integral is given in figure 4. Similar with the open string, the divergences in the $z \to 1$ and $z \to \infty$ limits are implied by the massless $t, u$ poles.

Using the crossing symmetry, we find the closed string celestial amplitudes for the other two kinematics

$$\mathbf{Im}\,\Psi_{\text{closed}}^{13\leftrightarrow24}(\beta, z) = -\pi 2^{3-\beta} z^{\frac{\beta}{2}+2} \sum_{n=1}^{\infty} \frac{(\sqrt{n})^{\beta-2}}{2} \left[ \frac{\left(\frac{n}{z-1}+1\right)_{n-1}}{n!} \right]^2,$$

$$\mathbf{Im}\,\Psi_{\text{closed}}^{14\leftrightarrow23}(\beta, z) = -\pi 2^{3-\beta} (-z)^{\frac{\beta}{2}+2} (1-z)^{-\frac{\beta}{2}} \sum_{n=1}^{\infty} \frac{(\sqrt{n})^{\beta-2}}{2} \left[ \frac{\left(\frac{nz}{1-z}+1\right)_{n-1}}{n!} \right]^2.$$
$$\tag{4.19}$$

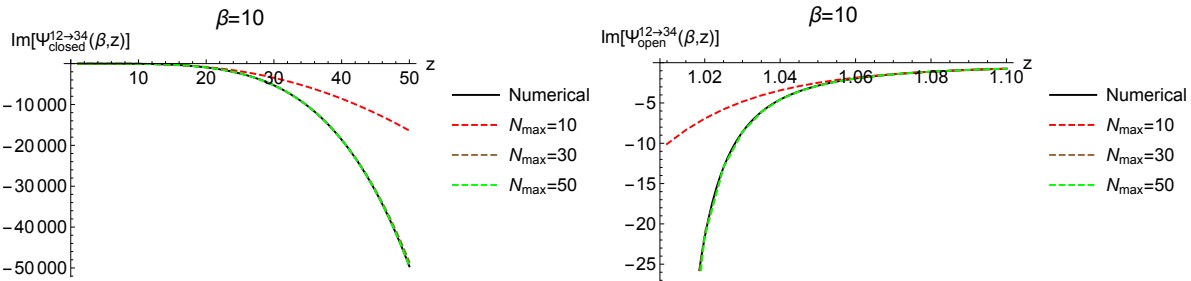

**Figure 4.** Here we present the comparison of numerical evaluation of Mellin integral and residue sum formula eq.(4.18) for the imaginary part of celestial closed string amplitude. LHS and RHS shows the behavior of $\mathbf{Im}\,\Psi_{\text{closed}}^{12\leftrightarrow24}(\beta, z)$ in $z \to \infty$ and $z \to 1$

## 5  Analytic continuation in $z$

As reviewed in sec.2, the celestial four-point amplitude consists of three different functions defined in three distinct regions on the real circle of the complex sphere. One can ask if the

three functions can be analytically connected to each other. Consider the simple massive scalar exchange example, and compare eq.(2.33) with

$$
\Psi^{13\leftrightarrow24}(\beta,z) = \frac{\pi g^2}{\sin\frac{\pi\beta}{2}} \left(\frac{m}{2}\right)^{\beta-2} z^2 \left[1 + e^{\frac{1}{2}\pi i\beta} z^{\frac{\beta}{2}} + \left(\frac{z}{1-z}\right)^{\frac{\beta}{2}}\right] \qquad (1 \geq z \geq 0),
$$

$$
\Psi^{14\leftrightarrow23}(\beta,z) = \frac{\pi g^2}{\sin\frac{\pi\beta}{2}} \left(\frac{m}{2}\right)^{\beta-2} z^2 \left[1 + (-z)^{\frac{\beta}{2}} + e^{\frac{1}{2}\pi i\beta} \left(\frac{-z}{1-z}\right)^{\frac{\beta}{2}}\right] \qquad (0 \geq z).
$$

(5.1)

It is straight forward to see that the functions cannot be analytically related.

There are, however, reasons we might want to consider the analytic continuation of a celestial amplitude from the physical region as specified in Table 1 to the unphysical ones. Recall that the presence of bulk factorization is imprinted in the imaginary part of of celestial sphere, where in each kinematic region only the channel with the physical threshold is projected. Thus for fixed $z$, to access the information of all factorization channels through the imaginary part, we will need to analytic continue $\Psi$ of other kinematic configurations to their unphysical regions. Such analytic continuation will be important when we consider dispersion relations for the celestial amplitude in Section 6.

## 5.1 The scalar example

Before turning to the general analysis, we will use the massive scalar exchange example to illustrate the last point. To continue $\Psi_{\mathrm{scalar}}^{12\leftrightarrow34}(\beta,z)$ outside the physical region $z > 1$, we need to be careful about potential monodromies as we continue across the boundaries at $z = 1$ and $z = \infty$. Continuing across $z = 1$ by $z - 1 \to e^{-\pi i}(1 - z)$ and across $z = \infty$ by $z \to e^{\pi i}(-z)$, we obtain

$$
\Psi_{\mathrm{scalar}}^{12\leftrightarrow34}(\beta,z) = \frac{\pi g^2}{\sin\frac{\pi\beta}{2}} \left(\frac{m}{2}\right)^{\beta-2} z^2 \begin{cases} e^{\frac{1}{2}\pi i\beta} + z^{\frac{\beta}{2}} + e^{\frac{1}{2}\pi i\beta}\left(\frac{z}{1-z}\right)^{\frac{\beta}{2}} & (1 > z > 0), \\ e^{\frac{1}{2}\pi i\beta} + e^{\frac{1}{2}\pi i\beta}(-z)^{\frac{\beta}{2}} + \left(\frac{z}{z-1}\right)^{\frac{\beta}{2}} & (0 > z). \end{cases}
$$

(5.2)

The imaginary part of $\Psi_{\mathrm{scalar}}^{12\leftrightarrow34}(\beta,z)$ in the new (unphysical) regions are

$$
\mathbf{Im}\, \Psi_{\mathrm{scalar}}^{12\leftrightarrow34}(\beta,z) = \pi g^2 \left(\frac{m}{2}\right)^{\beta-2} z^2 \begin{cases} 1 + \left(\frac{z}{1-z}\right)^{\frac{\beta}{2}} & (1 > z > 0), \\ 1 + (-z)^{\frac{\beta}{2}} & (0 > z), \end{cases}
$$

$$
= 2^{3-\beta}\pi z^2 g^2 \begin{cases} \underset{\omega\to m,\,\sqrt{\frac{z}{1-z}}m}{\mathbf{Res}} \left[\omega^{\beta-1} T_{\mathrm{scalar}}^{12\leftrightarrow34}(\omega,z)\right] & (1 > z > 0), \\ \underset{\omega\to m,\,\sqrt{-zm}}{\mathbf{Res}} \left[\omega^{\beta-1} T_{\mathrm{scalar}}^{12\leftrightarrow34}(\omega,z)\right] & (0 > z). \end{cases}
$$

(5.3)

We see that the analytically continued result receives contribution from the $s$- and $t$-channel poles in $1 > z > 0$ (the "$u$"-region), and $s$- and $u$-channel poles in $0 > z$ (the "$t$"-region). Note that the analytic continuation prescription here is uniquely fixed by demanding that the analytic continued amplitudes match irrespective of which physical region it originated

from. More precisely,

$$
\begin{aligned}
\Psi^{13\leftrightarrow24}_{\text{scalar}}(\beta, z) &\propto \Psi^{14\leftrightarrow23}_{\text{scalar}}(\beta, z) \quad (z > 1)\,, \\
\Psi^{14\leftrightarrow23}_{\text{scalar}}(\beta, z) &\propto \Psi^{12\leftrightarrow34}_{\text{scalar}}(\beta, z) \quad (1 > z > 0)\,, \\
\Psi^{12\leftrightarrow34}_{\text{scalar}}(\beta, z) &\propto \Psi^{13\leftrightarrow24}_{\text{scalar}}(\beta, z) \quad (0 > z)\,,
\end{aligned}
\tag{5.4}
$$

where the proportionality coefficients being $z$-independent phases. Explicit comparison fixes the phase to be $(1, e^{-\frac{1}{2}i\pi\beta}, e^{\frac{1}{2}i\pi\beta})$ respectively.

## 5.2 General analysis

Having gone through the scalar exercise, we are now ready to consider the continuation for general amplitudes. At the level of the integrand, the analytic continuation on $z$ corresponds to the analytic continuation on the Mandelstam variables from the physical to the "unphysical" kinematic regions, as shown in Table 3:

| | $12 \leftrightarrow 34$ | $13 \leftrightarrow 24$ | $14 \leftrightarrow 23$ |
|---|---|---|---|
| $z > 1$ | $s > 0 > u, t$ | $u, t > 0 > s$ | $u, t > 0 > s$ |
| $1 > z > 0$ | $s, t > 0 > u$ | $u > 0 > s, t$ | $s, t > 0 > u$ |
| $0 > z$ | $s, u > 0 > t$ | $s, u > 0 > t$ | $t > 0 > s, u$ |

**Table 3**. The physical and "unphysical" kinematic regions.

The physical regions are those that have only one positive Mandelstam, sitting on the diagonal entries. Take $\Psi^{12\leftrightarrow34}$ as an example. The analytic continuation to unphysical regions in $z$, say $0 > z$, leads to an unphysical kinematic setup $(s, u > 0 > t)$ and is distinct from $\Psi^{14\leftrightarrow23}$ originated from the physical setup $t > 0 > s, u$. Thus it is natural that the three distinct functions on the equator cannot be analytically continued to each other. On the other hand, continuing $\Psi^{13\leftrightarrow24}$ and $\Psi^{14\leftrightarrow23}$ to their common unphysical region, i.e. $z > 1$, do yield the same Mandelstam region $(u, t > 0 > s)$ and it is natural to identify their analytic continuation. The precise phase can be read off from known amplitudes. Thus we impose the following conditions on the analytic continued celestial amplitudes

$$
\begin{aligned}
\Psi^{13\leftrightarrow24}(\beta, z) &= \Psi^{14\leftrightarrow23}(\beta, z) & (z > 1)\,, \\
\Psi^{14\leftrightarrow23}(\beta, z) &= e^{-\frac{1}{2}i\pi\beta}\Psi^{12\leftrightarrow34}(\beta, z) & (1 > z > 0)\,, \\
\Psi^{12\leftrightarrow34}(\beta, z) &= e^{\frac{1}{2}i\pi\beta}\Psi^{13\leftrightarrow24}(\beta, z) & (0 > z)\,,
\end{aligned}
\tag{5.5}
$$

where the phases $e^{\pm\frac{1}{2}i\pi\beta}$ are due to analytic continuing the prefactor $B(\beta, z)$ in (2.24). These conditions could be viewed as the crossing symmetry for celestial amplitudes.[8] Indeed they are satisfied by the massive scalar exchange.

Note that to analytic continue, we need to ensure that the celestial amplitude exists along the path of continuation. More precisely, the celestial amplitude $\Psi(\beta, z)$ given by

---

[8]We thank the anonymous SciPost referee for emphasizing this point.

the integral of the flat-space amplitude $T(\omega, z)$ must be convergent along the path. As an example, consider continuing $\Psi^{12\leftrightarrow34}$ from its physical region $z > 1$ to $1 > z > 0$. More explicitly, our path of continuation is given by

$$
\begin{aligned}
P_1 : \;& \text{from any } z > 1 \text{ to } z = 1 + \epsilon \,, \\
P_2 : \;& z = 1 + \epsilon e^{-i\phi} \text{ for } \phi \text{ from } 0 \text{ to } \pi \,, \\
P_3 : \;& \text{from } z = 1 - \epsilon \text{ to any } z \in (0,1) \,,
\end{aligned}
\tag{5.6}
$$

where $\epsilon > 0$ is a small number. Previously in Section 3.1, we've discussed that the existence of the celestial amplitude requires $T^{12\leftrightarrow34}(\omega, z)$ to vanish as $\omega \to \infty$ in the physical region $z > 1$. We have further assumed that the convergence property of $T^{12\leftrightarrow34}(\omega, z)$ can be extended onto the complex $\omega$ plane with a finite range of argument, i.e. $T^{12\leftrightarrow34}(\omega, z) \to 0$ as $|\omega| \to \infty$ with the argument

$$
\begin{aligned}
\arg \omega \in \Theta^{12\leftrightarrow34}(z) \equiv \big(&- \theta_c^{12\leftrightarrow34}(z), \theta_c^{12\leftrightarrow34}(z)\big) \\
&\cup \big(\pi - \theta_c^{12\leftrightarrow34}(z), \pi + \theta_c^{12\leftrightarrow34}(z)\big) \quad \text{for} \quad z > 1 \,,
\end{aligned}
\tag{5.7}
$$

where we have emphasized the $z$ dependence by writing $\theta_c^{12\leftrightarrow34}(z)$. Now, let us consider the region $1 > z > 0$. Using the identity $T^{12\leftrightarrow34}(\omega, z) = T^{13\leftrightarrow24}(\frac{i\omega}{\sqrt{z}}, z)$, we find that $T^{12\leftrightarrow34}(\omega, z)$ vanishes as $|\omega| \to \infty$ with the argument

$$
\begin{aligned}
\arg \omega \in \Theta^{12\leftrightarrow34}(z) \equiv \Big(&\frac{\pi}{2} - \theta_c^{13\leftrightarrow24}(z), \frac{\pi}{2} + \theta_c^{13\leftrightarrow24}(z)\Big) \\
&\cup \Big(- \frac{\pi}{2} - \theta_c^{13\leftrightarrow24}(z), -\frac{\pi}{2} + \theta_c^{13\leftrightarrow24}(z)\Big) \quad \text{for} \quad 1 > z > 0 \,.
\end{aligned}
\tag{5.8}
$$

In general, the region (5.8) does not contain the angle $\arg \omega = 0$. Hence, we need to deform the integration contour continuously along the way of the analytic continuation $P_2$ in (5.6). For such continuous contour deformation to exist, the convergent region $\Theta(z)$ should exist and continuously move from (5.7) to (5.8) when going along the path $P_2$. This is illustrated in figure 5, where the region $\Theta(z)$ for $z > 1$ ($1 > z > 0$) is drawn as the blue (pink) region. For the massive scalar exchange the amplitude is convergent for all $\arg \omega \in [0, 2\pi)$, and one do not need to deform the integration contour. For string theory amplitudes, the blue region ($\Theta^{12\leftrightarrow34}(z)$ for $z > 1$) is computed in sec.4 and the pink region ($\Theta^{12\leftrightarrow34}(z)$ for $1 > z > 0$) can be obtained by the formula (5.8) using the $\theta_c^{13\leftrightarrow24}$ given in Table 2. As we will see in sec.C the convergent region $\Theta(z)$ deforms continuously from the blue to the pink regions when going along the path $P_2$.

With this understanding, we can now carry out the analytic continuation. Begin with $\Psi^{12\leftrightarrow34}$ in the physical region $z > 1$ defined by the Mellin integral (2.24), which is an integral along the contour $I_1$ in figure 5. When going along the path (5.6), the integration contour should deform accordingly, such that it is always inside the corresponding angular region $\Theta(z)$. When arriving the destination $z \in (0,1)$, the integration contour becomes $-I_2$ in figure 5. Finally, under the analytic continuation, the $t$-channel poles (represented by the red dots in figure 5) are rotated from the imaginary axis to the real axis, while the $s$- and $u$-channel poles (represented by the black and blue dots in figure 5) remain near the real and the imaginary axes, respectively.

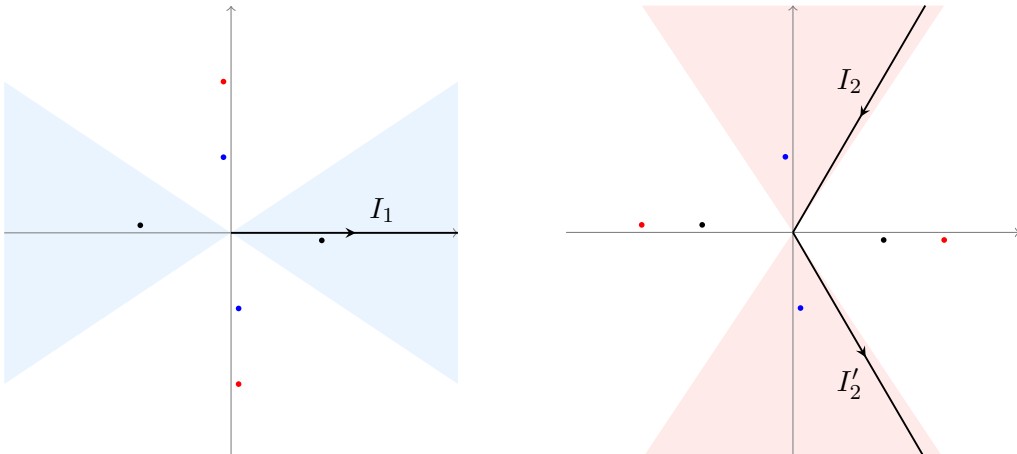

**Figure 5**. The blue (pink) regions is the convergent angular region $\Theta(z)$ for $z = 1 + \epsilon > 1$ ($z = 1 - \epsilon < 1$). The black, blue or red dot is a representative for the $s$-, $t$- or $u$-channel poles at $\omega = \pm m$, $\omega = \pm i\sqrt{\frac{z}{z-1}}m_i$ or $\omega = \pm i\sqrt{z}m_i$, respectively.

Now, let us consider the imaginary part of the celestial amplitude $\mathbf{Im}\,\Psi^{12\leftrightarrow34}(\beta, z)$ after analytically continuing to $1 > z > 0$. Similar to the discussion in Section 3.1, it can be computed by the integral along the contour $I_2 + I_2'$ as shown in figure 5. However, in this case, we cannot close the contour by adding the $I_3'$ piece as in figure 1, because the $I_3'$ contour extends outside the pink angular region, and the contour integral diverges. When applied to the case of the massive scalar exchange amplitude studied in Section 2.3, we have a better asymptotic property that

$$\Theta(z) = [0, 2\pi) \quad \forall z \in \mathbb{C}\,. \tag{5.9}$$

In such case, $\mathbf{Im}\,\Psi^{12\leftrightarrow34}(\beta, z)$ in the region $1 > z > 0$ can also be computed by the fan-like contour $I_2 + I_2' + I_3'$, which picks up the residues and discontinuities of the poles and branch cuts inside the contour. The result is

$$
\begin{aligned}
&\mathbf{Im}\,\Psi^{12\leftrightarrow34}_{>\Lambda}(\beta, z) \\
&= -B^{12\leftrightarrow34}(z)\pi \sum_i \mathop{\mathbf{Res}}_{\omega \to m_i,\, \sqrt{\frac{z}{1-z}}m_i} \left[\omega^{\beta-1}T^{12\leftrightarrow34}(\omega, z)\right] \\
&\quad + \left(\int_M^\infty + \int_{\sqrt{\frac{z}{1-z}}M}^\infty\right) \mathrm{d}\omega\,\omega^{\beta-1}\,\mathbf{Disc}\left[T^{12\leftrightarrow34}(\omega, z)\right] \quad (1 > z > 0)\,.
\end{aligned} \tag{5.10}
$$

We see that the imaginary part of (5.2) indeed agrees with the formula (5.10).

## 6 Celestial dispersion relation

The analyticity properties of a scattering amplitude $T(s, t)$ can be summarized by dispersion relations. When the Mandelstam variable $u = u_0$ is fixed with the absolute value

$|u_0|$ smaller than the multi-particle production threshold $M$, the dispersion relation for the scattering amplitude reads[9]

$$T(s,t) = -\sum_i \frac{g_i^2}{s - m_i^2} - \sum_i \frac{g_i^2}{t - m_i^2}$$
$$+ \frac{1}{\pi} \int_{M^2}^\infty ds' \frac{\mathbf{Disc}_s [T(s',t)]}{s' - s} + \frac{1}{\pi} \int_{M^2}^\infty dt' \frac{\mathbf{Disc}_t [T(s,t')]}{t' - t}, \tag{6.1}$$

where the discontinuity $\mathbf{Disc}_s[T(s,t)]$ is defined by

$$\mathbf{Disc}_s[T(s,t)] = \frac{1}{2i} \left( T(s + i\epsilon, t) - T(s - i\epsilon, t) \right), \tag{6.2}$$

and similarly for $\mathbf{Disc}_t[T(s,t)]$. By unitarity, they are positively expandable on appropriate orthogonal polynomials. Along the line of S-matrix bootstrap, the dispersion relations have been used recently to constrain the EFT coefficients – the expansion coefficients of the amplitude $T(s,t)$ at $s = u = t = 0$ [23, 25–28].

On the celestial sphere, on one hand, as we have seen in Section 3, the poles and discontinuities that appear on the right hand side of the dispersion relation (6.1) contribute to the imaginary part of the celestial amplitude. On the other hand, as elucidated in [21] and also discussed in Section 2.2, the EFT coefficients show up as the residues of the poles in the celestial amplitude on the complex $\beta$-plane at negative even integers. Therefore, it is natural to expect a relation between the residues of the poles at $\beta = -2n$ for $n \in \mathbb{Z}_{\leq 0}$ and the imaginary part of the celestial amplitude, given by the projection of the dispersion relation (6.1) on the celestial sphere. However, the dispersion relation (6.1) cannot be directly translated to the celestial sphere by the Mellin transform (2.24), since for generic $z$, the $\omega$-integral extends to the region where all the Mandelstam variables are large and the dispersion relation (6.1) does not hold. Keeping $u = u_0$ fixed and finite corresponds to the OPE limit (colinear limit) of the celestial amplitude, for example, the $z \to \infty$ limit in the $12 \leftrightarrow 34$ kinematics.

From the celestial amplitude of the massive scalar exchange (2.33), we observe the following relation between the residues of the poles at negative even integer $\beta$ and the imaginary part of the celestial amplitude

$$\frac{\pi}{2} \operatorname*{\mathbf{Res}}_{\beta \to -2n} \left[ \Psi_{\text{scalar}}^{12 \leftrightarrow 34}(\beta, z) \right] = \mathbf{Im} \left[ \Psi_{\text{scalar}}^{12 \leftrightarrow 34}(\beta, z) + (-1)^n \Psi_{\text{scalar}}^{13 \leftrightarrow 24}(\beta, z) \right] \Big|_{\beta \to -2n} \quad (z \geq 1), \tag{6.3}$$

and similar relations in the other regions. Importantly, the celestial amplitude $\Psi_{\text{scalar}}^{13 \leftrightarrow 24}(-2n, z)$ on the right hand side is given by the analytic continuation from the physical defining region $1 > z > 0$ to the region $z > 1$, as discussed in Section 5.1. Note that we could have chosen the analytic continuation of $\Psi_{\text{scalar}}^{14 \leftrightarrow 23}$ instead, since from eq.(5.5), $\Psi_{\text{scalar}}^{13 \leftrightarrow 24}(\beta, z) = \Psi_{\text{scalar}}^{14 \leftrightarrow 23}(\beta, z)$ when $z \geq 1$.

---

[9]We assume the asymptotic behavior of the amplitude as $T(s, -s - u_0) \to 0$ when $|s| \to \infty$. For amplitudes with worse asymptotic behavior like $T(s, -s - u_0) \to |s|^N$ when $|s| \to \infty$, similar dispersion relations can be written down with $N$ unknown coefficients known as subtraction constants.

More general celestial amplitudes satisfy a similar celestial dispersion relation,

$$\frac{\pi}{2} \operatorname*{Res}_{\beta \to -2n} \left[ \Psi^{12\leftrightarrow34}(\beta, z) \right] = \mathbf{Im} \left[ \Psi^{12\leftrightarrow34}(\beta, z) + (-1)^n \Psi^{13\leftrightarrow24}(\beta, z) \right] \Big|_{\beta \to -2n} \quad (z \geq 1).$$

(6.4)

In the above, we take $n \geq 1$ if gravity is non-dynamical and $n > 1$ when graviton exchange are involved. The celestial amplitude $\Psi^{13\leftrightarrow24}(-2n, z)$ on the right hand side of (6.4) is given by the analytic continuation from the physical region $1 \geq z \geq 0$ to the region $z \geq 1$, as discussed in Section 5.

Let us now derive (6.4). First we note that the original Mellin integral (2.24) can be written as a contour integral along the negative real axes:

$$\Psi^{12\leftrightarrow34}(\beta, z) = \frac{1}{2i} \csc(\pi\beta) B^{12\leftrightarrow34}(z) \oint_{\mathcal{C}_{(-\infty,0]}} d\omega \, \omega^{\beta-1} T^{12\leftrightarrow34}(\omega, z),$$

(6.5)

where the contour $\mathcal{C}_{(-\infty,0]}$ is a thin counterclockwise contour along the negative real axes that picks up the discontinuity across the branch cut associated with the $\omega^{\beta-1}$ factor, as shown in the left of figure 6. Note that we've used $i\epsilon$ to push the cuts associated with $T$ on to the complex $\omega$ plane. This identity can be understood by noting that

$$\mathbf{Disc}[\omega^{\beta-1}] = \frac{1}{2i} \left[ (|\omega|e^{i\pi})^{\beta-1} - (|\omega|e^{-i\pi})^{\beta-1} \right] = -\sin(\pi\beta)|\omega|^{\beta-1} \quad \text{when} \quad \omega \in (-\infty, 0].$$

(6.6)

via. change of variable, we recover the original Mellin integral. This formula makes the pole structure of the celestial amplitude manifest as the $\csc(\pi\beta)$ pre-factor, and the contour integral along $\mathcal{C}_{(-\infty,0]}$ converges on the entire $\beta$-plane. With this formula, the celestial amplitude $\Psi(\beta, z)$ can be analytically continued to $\beta \leq 0$. The pre-factor $\csc(\pi\beta)$ has simple poles at $\beta \in \mathbb{Z}_{\leq 0}$, for which $\omega^{\beta-1}$ yields a (multi)pole at $\omega = 0$. Thus when computing the residue for $\beta \in \mathbb{Z}_{\leq 0}$, the contour $\mathcal{C}_{(-\infty,0]}$ is contracted to a small circle $\mathcal{C}_0$ centered at the origin. We then arrive at[10]

$$\operatorname*{Res}_{\beta \to -2n} \left[ \Psi^{12\leftrightarrow34}(\beta, z) \right] = \frac{1}{2\pi i} B(z) \oint_{\mathcal{C}_0} d\omega \, \omega^{-2n-1} T^{12\leftrightarrow34}(\omega, z).$$

(6.7)

To proceed, let us impose the following two assumptions on $T^{12\leftrightarrow34}(\omega, z)$.

1. **Maximal analyticity**: The only poles and branch cuts of $T^{12\leftrightarrow34}(\omega, z)$ are located near the real or imaginary axis on the $\omega$-plane as shown in figure 6. They correspond to the exchange of single or multi-particle states.

2. **Boundedness**: $T^{12\leftrightarrow34}(\omega, z)$ is bounded for some finite extension onto the the complex $\omega$, i.e. $T^{12\leftrightarrow34}(\omega, z) \to 0$ as $|\omega| \to \infty$ with the argument

$$\arg \omega \in \left( -\theta_c^{12\leftrightarrow34}, \theta_c^{12\leftrightarrow34} \right),$$

(6.8)

---

[10]The residues of the poles at $\beta = -2n-1$ all equal to zero, because $T(\omega, z)$ is an even function in $\omega$.

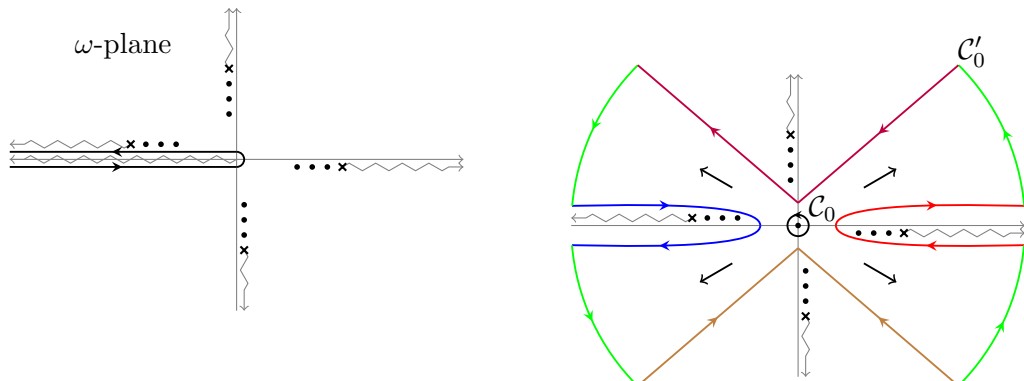

**Figure 6**. **Left:** The contour $\mathcal{C}_{(-\infty,0]}$. **Right:** Pulling the contour $\mathcal{C}_0$ to $\mathcal{C}_0'$, which further decomposes into the red contour $\mathcal{D}_0$, the blue contour $\mathcal{D}_1$, the purple contour $\mathcal{D}_2$, the brown contour $\mathcal{D}_3$, and finally the green contours.

Now, let us pull the contour $\mathcal{C}_0$ to the contour $\mathcal{C}_0'$ as shown in figure 6, where the contour is now fan-shaped with the extension angle determined by $\theta_c^{12\leftrightarrow34}$. Due to the boundedness assumption, the arcs of the contour $\mathcal{C}_0'$ at infinite (the green contours) do not contribute to the integral.

The remaining part of the contour $\mathcal{C}_0'$ decomposes into four pieces: the red contour $\mathcal{D}_0$, the blue contour $\mathcal{D}_1$, the purple contour $\mathcal{D}_2$, and the brown contour $\mathcal{D}_3$. The contour integral along $\mathcal{D}_1$ can be simply related to the contour integral along $\mathcal{D}_0$ and using (3.5), we get

$$
\begin{aligned}
\frac{\pi z^2}{2\pi i}\int_{\mathcal{D}_1} d\omega\, \omega^{-2n-1}T^{12\to34}(\omega,z) &= e^{-2n\pi i}\frac{\pi z^2}{2\pi i}\int_{\mathcal{D}_0} d\omega\, \omega^{-2n-1}T^{12\to34}(\omega,z)\\
&= \frac{1}{\pi}\mathbf{Im}[\Psi^{12\to34}(\beta,z)]\,.
\end{aligned}
\tag{6.9}
$$

Next, performing the change of variable $\omega = i\sqrt{z}\omega'$, we find:

$$
\frac{\pi z^2}{2\pi i}\int_{\mathcal{D}_2+\mathcal{D}_3} d\omega\, \omega^{-2n-1}T^{12\to34}(\omega,z) = \frac{(-1)^n}{2i}z^{2-n}\int_{\mathcal{D}_2'+\mathcal{D}_3'} d\omega'(\omega')^{-2n-1}T^{13\to24}(\omega',z)\,,
\tag{6.10}
$$

where $\mathcal{D}_2'$ and $\mathcal{D}_3'$ are the contours given by rotating the contours $\mathcal{D}_2$ and $\mathcal{D}_3$ clockwise around origin by $\pi/2$. The contour integral along $\mathcal{D}_3'$ can be simply related to the contour integral along $\mathcal{D}_2'$, which equals to the $I_2 + I_2'$ in Figure 5 contour that computes the imaginary part of the analytic continued celestial amplitude. More explicitly, we have

$$
\begin{aligned}
\frac{z^{2-n}\pi}{2i}\int_{\mathcal{D}_3'} d\omega'(\omega')^{-2n-1}T^{13\to24}(\omega,z) &= e^{-2\pi ni}\frac{z^{2-n}\pi}{2i}\int_{\mathcal{D}_2'} d\omega'(\omega')^{-2n-1}T^{13\to24}(\omega,z)\\
&= \mathbf{Im}[\Psi^{13\to24}(\beta,z)]\,.
\end{aligned}
\tag{6.11}
$$

Putting everything together, we find the celestial dispersion relation (6.4).

Note that for the amplitude of the massive scalar exchange (2.30), since the amplitude vanishes in all direction, we can instead deform the contour to $\mathcal{C}_0'$ in Figure 7. Therefore,

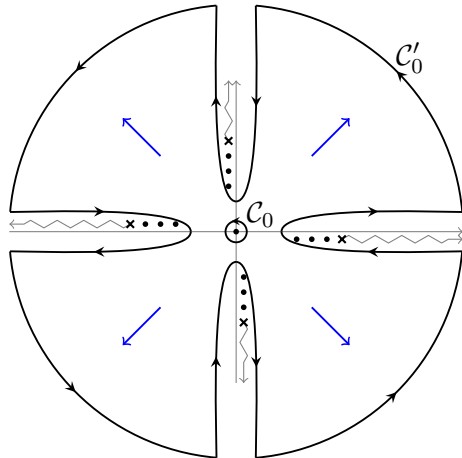

**Figure 7**. Contour deformation from $\mathcal{C}_0$ to $\mathcal{C}_0'$.

the integral picks up contributions from the poles and branch cuts near the real axis and gives the imaginary part of $\Psi^{12\leftrightarrow34}(\beta, z)$ by (3.5). The contribution from the imaginary axis is

$$
- B(z) \left\{ \pi \sum_i \mathop{\mathbf{Res}}_{\omega \to i\sqrt{z}m_i,\, i\sqrt{\frac{z}{z-1}}m_i} \left[ \omega^{-2n-1} T^{12\leftrightarrow34}(\omega, z) \right] \right.
$$
$$
\left. + \left( \int_{i\sqrt{z}M}^{i\infty} + \int_{i\sqrt{\frac{z}{z-1}}M}^{i\infty} \right) \mathrm{d}\omega\, \omega^{-2n-1} \mathbf{Disc}\left[ T^{12\leftrightarrow34}(\omega, z) \right] \right\}, \tag{6.12}
$$

which equals to $\mathbf{Im}\left[(-1)^n \Psi^{13\leftrightarrow24}(-2n, z)\right]$ or $\mathbf{Im}\left[(-1)^n \Psi^{14\leftrightarrow23}(-2n, z)\right]$, by the formulae similar to (5.10) with the appropriate changes of the integration variable, $\omega \to i\sqrt{z}\omega$ or $\omega \to i\sqrt{\frac{z}{z-1}}\omega$.

## 6.1 Celestial dispersion relation in string theory

Let us do the numerical check of our conjecture (6.4) for the celestial dispersion relation, by considering the open and closed string celestial amplitudes. Specifically, we will give the analytic form of the residues at $\beta = -10$ and $\beta = -12$ poles and numerically evaluate $\mathbf{Im}\,\Psi^{12\leftrightarrow34}$ and $\mathbf{Im}\,\Psi^{13\leftrightarrow24}$ through the fan-like integrals as indicated in (6.9) and (6.11). Finally, we will plot the results above and show they indeed match.

- Numeric results of $\beta = -10$
  The analytic forms of residues at the $\beta = -10$ poles of open and closed string celestial

amplitude are,

$$
\frac{\pi}{2} \operatorname*{\mathbf{Res}}_{\beta \to -10} \left[ \Psi_{\text{open}}^{12 \leftrightarrow 34}(\beta, z) \right]
$$

$$
= \frac{512\pi}{45 z^3} \left[ -\pi^4 z(-1+z)(4+7z(-1+z))\zeta(3) - 60\pi^2 z(-1+z)(1+z(-1+z))\zeta(5) \right.
$$
$$
\left. -360(1+z(-1+z))^2 \zeta(7) \right] ,
$$

$$
\frac{\pi}{2} \operatorname*{\mathbf{Res}}_{\beta \to -10} \left[ \Psi_{\text{closed}}^{12 \leftrightarrow 34}(\beta, z) \right] = - \frac{16384\pi(-1+z)\left[1+z(-1+z)\right]\zeta(3)\zeta(5)}{z^2}.
$$

(6.13)

where $\zeta(x)$ is the Riemann zeta function. The comparison with the RHS of (6.4) is shown in Figure 8.

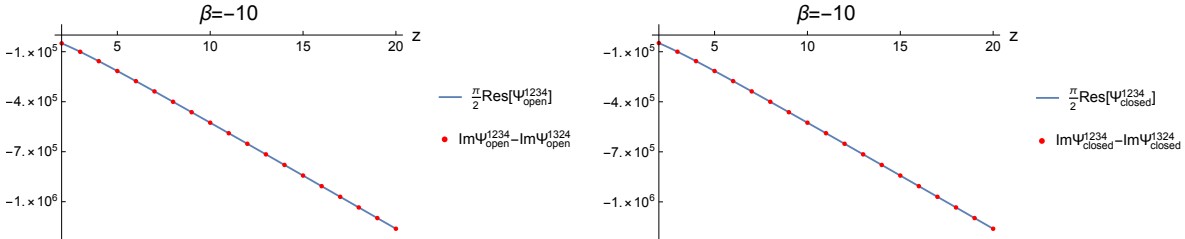

**Figure 8**. The comparison between the analytic formula for the residue at $\beta = -10$ and the numerical evaluation of the imaginary part for the open and closed string celestial amplitudes.

- Numeric results of $\beta = -12$

  The analytic forms of residues at the $\beta = -12$ poles of open and closed string celestial amplitude are,

$$
\frac{\pi}{2} \operatorname*{\mathbf{Res}}_{\beta \to -12} \left[ \Psi_{\text{open}}^{12 \leftrightarrow 34}(\beta, z) \right]
$$

$$
= \frac{128\pi}{14175 z^4} \left[ \pi^8 \left( -192 - z(-1+z)(912 + z(-1+z)(1256 + 381z(-1+z)))) \right) \right.
$$
$$
\left. -151200\pi^2 \zeta(3)^2 (-1+z)^2 z^2 - 1814400 \zeta(3)\zeta(5) z(-1+z)(1+z(-1+z)) \right] ,
$$

$$
\frac{\pi}{2} \operatorname*{\mathbf{Res}}_{\beta \to -12} \left[ \Psi_{\text{closed}}^{12 \leftrightarrow 34}(\beta, z) \right]
$$

$$
= \frac{256\pi}{945 z^4} \left[ 3\psi^{(8)}(1) + 9z(-1+z) \left( \psi^{(8)}(1) - 4480z(-1+z)(2\zeta(3)^3 + (10 + 3z(-1+z)\zeta(9)))) \right) \right] ,
$$

(6.14)

where $\psi^{(n)}(z)$ is the polygamma function. The comparison is given in Figure 9.

## 6.2 OPE limit

In the celestial dispersion relation, on the right hand side, the first term $\mathbf{Im}\,\Psi^{12 \leftrightarrow 34}(\beta, z)$ equals to a sum over the contributions from the $s$-channel exchange as discussed in Section 3. However, in general, the second term $\mathbf{Im}\,\Psi^{13 \leftrightarrow 24}(\beta, z)$ cannot be expressed as a sum over contributions from exchange process. The obstruction is due to the fact that in general the amplitude $T^{13 \leftrightarrow 24}(\omega, z)$ does not converge in the $\omega \to \infty$ limit when $z > 1$,

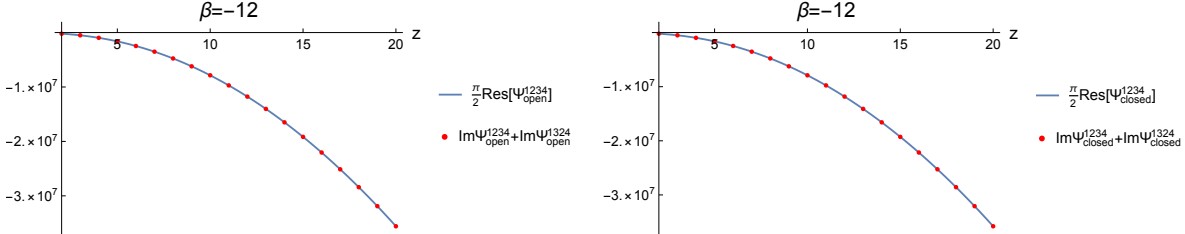

**Figure 9**. The comparison between the analytic formula for the residue at $\beta = -10$ and the numerical evaluation of the imaginary part for the open and closed string celestial amplitudes.

and thereby we cannot closed the $I_2 + I_2'$ contour in Figure 5.[11] This is exactly the case for the open and closed string amplitudes, as we have seen in the above analysis that $\mathbf{Im}\,\Psi^{13\to24}_{\text{open/closed}}(\beta, z)$ cannot be expressed as a sum over residues.

However, the high energy behaviors of the string scattering amplitudes become nicer in the OPE limit (co-linear limit). This would allow us to close the $I_2 + I_2'$ contour for the leading terms in the OPE. In the following analysis, we choose to work with $\mathbf{Im}\,\Psi^{14\to23}_{\text{open/closed}}(\beta, z)$ by noting the relations (5.5). Let us expand the open string amplitude (4.5) in the OPE limit $z \to \infty$ as

$$T^{14\leftrightarrow23}_{\text{open}}(\omega, z) = T_{\text{open}}\left(\frac{z}{1-z}\omega^2, \omega^2\right) = \sum_{m=0}^{\infty} T^{14\leftrightarrow23,(m)}_{\text{open}}(\omega) z^{-m}. \tag{6.15}$$

From the explicit amplitude (4.5), we find that in the high energy limit $\omega \to \infty$ the expansion coefficients behave as

$$T^{14\leftrightarrow23,(m)}_{\text{open}}(\omega) \sim \omega^{2m-2}\csc(\pi\omega^2)\log^m(\omega). \tag{6.16}$$

The $I_2 + I_2'$ contour in Figure 5 can be closed at infinity when $\beta \leq -2m + 2$. Hence, the leading terms in the expansion of $\mathbf{Im}\Psi^{14\to23}_{\text{open}}(\beta, z)$ can be computed by a sum over the residue of the $t$-channel poles. For example, we have

$$\mathbf{Im}\left[\Psi^{14\to23}_{\text{open}}(\beta, z)\right]\Big|_{\beta\to0} = -\pi B^{14\to23}(z)\left[\sum_{n=1}^{\infty}\frac{(-1)^{n+1}}{2n^2} + \sum_{n=1}^{\infty}\frac{(-1)^{n+1}(\psi^{(0)}(n)+\gamma)}{2n}z^{-1} + \mathcal{O}(z^{-2})\right]$$

$$= -\pi B^{14\to23}(z)\left[\frac{\pi^2}{24} - \frac{1}{4}\log^2(2)z^{-1} + \mathcal{O}(z^{-2})\right]. \tag{6.17}$$

Similarly, the closed string amplitude (4.8) expanded in the OPE limit $z \to \infty$ gives

$$T^{14\leftrightarrow23}_{\text{closed}}(\omega, z) = T_{\text{closed}}\left(\frac{z}{1-z}\omega^2, \omega^2\right) = \sum_{m=-1}^{\infty} T^{14\leftrightarrow23,(m)}_{\text{closed}}(\omega) z^{-m}. \tag{6.18}$$

---

[11]The analysis around Figure 5 is for analytic continuing $T^{12\leftrightarrow34}(\omega, z)$ from $z > 1$ to $1 > z > 0$, but we can repeat the same analysis for analytic continuing $T^{13\leftrightarrow24}(\omega, z)$ from $1 > z > 0$ to $z > 1$, and the Figure 5 still applies upon reinterpreted the black, blue and red dots as representatives for the $u$-, $s$-, and $t$-channel poles.

In the high energy limit $\omega \to \infty$, we find that the expansion coefficients behave as

$$T_{\text{closed}}^{14\leftrightarrow23,(m)}(\omega) \sim \omega^{2m-4} \log^{m+1}(\omega).\tag{6.19}$$

The $I_2 + I_2'$ contour can be closed when $\beta < -2m + 4$. Again, the leading terms in the expansion of $\mathbf{Im}\Psi_{\text{closed}}^{14\to23}(\beta, z)$ can be computed by a sum over the residue of the $t$-channel poles. For example, we have

$$\begin{aligned}
\mathbf{Im}\left[\Psi_{\text{open}}^{14\to23}(\beta, z)\right]\Big|_{\beta\to0} &= -\pi B^{14\to23}(z)\left[\sum_{n=1}^{\infty}\frac{1}{2n^3} + \sum_{n=1}^{\infty}\frac{\psi^{(0)}(n) + \gamma}{n^2}z^{-1} + \mathcal{O}(z^{-2})\right]\\
&= -\pi B^{14\to23}(z)\left[\frac{\zeta(3)}{2} + \zeta(3)z^{-1} + \mathcal{O}(z^{-2})\right].
\end{aligned}\tag{6.20}$$

One can similarly compute the imaginary part of the open and closed string celestial amplitudes in the OPE limit for other non-positive integer $\beta$ and verify the celestial dispersion relation for the leading orders in the large $z$ expansion.

# 7 Conclusion and outlook

In this paper, we studied the relations between the bulk-locality, unitarity and analyticity of the celestial amplitudes.

1. We showed that the imaginary part of the celestial amplitude can be expressed as the integral along the fan-like contour (the right plot in Figure 1) that encloses the positive real axis on the complex plane of the center of mass energy $\omega$. The exchanges of single- and multi-particle states produce poles and branch cuts near the positive real axis and contribute to the fan-like contour integral. This demonstrates that the imaginary part of the celestial amplitude is given by bulk factorization singularities.

2. By unitarity, the imaginary part of the celestial amplitude can be positively expanded in the basis of Legendre or Jacobi polynomials for scalar or spinning particle amplitudes. The projection of these orthogonal polynomials onto the celestial sphere can be matched to the Poincaré partial waves that satisfy the massive Casimir equation of the Poincaré algebra.

3. The four-point celestial amplitude from three distinct physical kinematic configurations tiles the equator of the celestial sphere. On the boundary of each region that functions on the two sides are not continuously connected. Instead we studied the analytic continuation of the celestial amplitude from the physical regions (the intervals listed in Table 1) to the to the unphysical region.

4. Assuming specific high energy behaviour for the flat-space amplitude for complex scattering angles, we prove a celestial version of the dispersion relation (6.4). This allows us to relate the residues of the poles of the celestial amplitude at negative even integers on the complex $\beta$-plane, which is yields the EFT coefficients, to the imaginary part of the celestial amplitude and its analytic continuation.

5. These results are verified in the context of the open and closed string amplitudes.

In defining the fan contour, we've assumed that the amplitude is well behaved as $\omega \to \infty$ even as $\omega$ is continued on to the complex plane with a small angle. While we've shown that this is holds for scalar exchange and string theory amplitudes, it will be desirable to derive such behaviour from some general principle.

The celestial dispersion relation is reminiscent of the Zamolodchikov's recursive representation for the conformal blocks [29–31]. When tuning the conformal dimension of a primary operator to some special values, certain descendant operators become zero norm and decouple from the conformal multiplet. This phenomenon reflects on the conformal block as poles on the complex conformal dimension plane whose residues are the conformal blocks formed by the zero norm descendants. In our celestial dispersion relation (6.4), we see that the residue of the poles at negative even integer $\beta$ are the imaginary part of the celestial amplitude, which can be further expanded in terms of the Poincaré partial waves. It would be interesting to see if the origin of these poles can be traced by to the exchange of some zero norm states in the celestial CFT.

## Acknowledgments

We would like to thank Shu-Heng Shao and Yu-Chi Hou for enlightening discussions. Cm C would like to thank the hospitality of NTU high-energy theory group. Cm C is partly supported by National Key R&D Program of China (NO. 2020YFA0713000). Yt H, and Zx H is supported by MoST Grant No. 109-2112-M-002 -020 -MY33. Yt H is also supported by Golden Jade fellowship.

## A    Four-point helicity amplitude

Let us consider a four-point amplitude of massless particles with helicities $\ell_i$ for $i = 1, \cdots, 4$. The amplitude can be expressed as a function of the angle brackets $\langle ij \rangle$ and the square brackets $[ij]$. The Mandelstam variables are products of angle and square brackets, $s_{ij} = \langle ij \rangle [ji]$. Hence, we can instead choose to express the amplitude as a function of the angle bracket $\langle ij \rangle$ and $s, t$. There are relations among these variables. First, we have the Schouten identity

$$\langle 41 \rangle \langle 23 \rangle + \langle 42 \rangle \langle 31 \rangle + \langle 43 \rangle \langle 12 \rangle = 0 \,. \tag{A.1}$$

We also have relations from the momentum conservation

$$\frac{\langle 23 \rangle}{\langle 13 \rangle}(-s-t) + \frac{\langle 24 \rangle}{\langle 14 \rangle}t = 0 \,, \qquad \frac{\langle 12 \rangle}{\langle 32 \rangle}t + \frac{\langle 14 \rangle}{\langle 34 \rangle}s = 0 \,, \quad \frac{\langle 12 \rangle}{\langle 42 \rangle}(-s-t) + \frac{\langle 13 \rangle}{\langle 43 \rangle}s = 0 \,. \tag{A.2}$$

We can rewrite the left hand side of the above equation as a $3 \times 2$ matrix acting on the vector $(s, t)$, and the Schouten identity implies that all the second minors of matrix vanish. Hence, the relations (A.2) all linearly relate to each others.

Using (A.1) and (A.2), we can write the amplitude as a function of the variables

$$\langle 12\rangle , \quad \langle 13\rangle , \quad \langle 14\rangle , \quad \langle 34\rangle , \quad s , \quad t , \tag{A.3}$$

or equivalently as a function of the variables

$$\frac{z_{12}}{\bar{z}_{12}} = -\epsilon_1\epsilon_2 \frac{\langle 12\rangle^2}{s} , \quad \frac{z_{13}}{\bar{z}_{13}} = \epsilon_1\epsilon_3 \frac{\langle 13\rangle^2}{s+t} , \quad \frac{z_{14}}{\bar{z}_{14}} = -\epsilon_1\epsilon_4 \frac{\langle 14\rangle^2}{t} ,$$

$$\frac{z_{34}}{\bar{z}_{34}} = -\epsilon_3\epsilon_4 \frac{\langle 34\rangle^2}{s} , \quad s , \quad t . \tag{A.4}$$

Now, using the constraint from the $\mathrm{SL}(2,\mathbb{C})$ symmetry (2.8), we find

$$\mathcal{A}_{\ell_i}(\omega_i, z_i) = \left(\frac{z_{12}}{\bar{z}_{12}}\right)^{-\ell_2} \left(\frac{z_{13}}{\bar{z}_{13}}\right)^{\frac{-\ell_1+\ell_2-\ell_3+\ell_4}{2}} \left(\frac{z_{14}}{\bar{z}_{14}}\right)^{\frac{-\ell_1+\ell_2+\ell_3-\ell_4}{2}} \left(\frac{z_{34}}{\bar{z}_{34}}\right)^{\frac{\ell_1-\ell_2-\ell_3-\ell_4}{2}}$$

$$\times\, \delta^{(4)}(p_1+p_2+p_3+p_4)T(s,t)$$

$$= \delta^{(4)}(p_1+p_2+p_3+p_4) \frac{\left(\frac{z_{14}\bar{z}_{13}}{\bar{z}_{14}z_{13}}\right)^{\frac{\ell_3-\ell_4}{2}} \left(\frac{z_{24}\bar{z}_{14}}{\bar{z}_{24}z_{14}}\right)^{\frac{\ell_1-\ell_2}{2}}}{\left(\frac{z_{12}}{\bar{z}_{12}}\right)^{\frac{\ell_1+\ell_2}{2}} \left(\frac{z_{34}}{\bar{z}_{34}}\right)^{\frac{\ell_3+\ell_4}{2}}} T(s,t) , \tag{A.5}$$

where in the second equality we have used the identities (A.2).

## B  Poincaré generators on single particle states

The Poincaré generators $P^\mu$ and $M^{\mu\nu}$ acting on a massless single particle state $|\Delta, z, \ell\rangle$ in the conformal primary basis as

$$P^\mu|\Delta, z, \ell\rangle = \mathcal{P}^\mu|\Delta, z, \ell\rangle , \quad M^{\mu\nu}|\Delta, z, \ell\rangle = \mathcal{M}^{\mu\nu}|\Delta, z, \ell\rangle , \tag{B.1}$$

where $\mathcal{P}^\mu$ and $\mathcal{M}^{\mu\nu}$ are differential operators, whose explicit form are given by [12]

$$\mathcal{M}^{01} = \frac{i}{2}\left[(\bar{z}^2-1)\bar{\partial} + (z^2-1)\partial + 2(\bar{h}\bar{z}+hz)\right] ,$$

$$\mathcal{M}^{02} = -\frac{1}{2}\left[(\bar{z}^2+1)\bar{\partial} - (z^2+1)\partial + 2(\bar{h}\bar{z}-hz)\right] ,$$

$$\mathcal{M}^{03} = i(\bar{z}\bar{\partial} + z\partial + \bar{h} + h) ,$$

$$\mathcal{M}^{12} = -\bar{z}\bar{\partial} + z\partial - \bar{h} + h , \tag{B.2}$$

$$\mathcal{M}^{13} = \frac{i}{2}\left[(\bar{z}^2+1)\bar{\partial} + (z^2+1)\partial + 2(\bar{h}\bar{z}+hz)\right] ,$$

$$\mathcal{M}^{23} = -\frac{1}{2}\left[(\bar{z}^2-1)\bar{\partial} - (z^2-1)\partial + 2(\bar{h}\bar{z}-hz)\right] ,$$

and

$$\mathcal{P}^\mu = 2q^\mu e^{\partial_\Delta} . \tag{B.3}$$

## C   Analytic continuation of the string amplitudes

In this appendix, we apply the analytic continuation procedure in Section 5.2 to the open and closed string amplitudes. As discussed in Section 5.2, the integration contour of the celestial amplitude are being continuously deformed when going along the path (5.6). The integration contour should asymptote to the angle inside the convergent region $\Theta^{12\leftrightarrow34}(z)$. Hence, we need to ensure that the convergent region varies continuously along the path (5.6). Let us parametrize the string amplitudes (4.5) and (4.8) as

$$s = \omega^2, \quad t = -\frac{z-1}{z}\omega^2, \quad \omega = re^{i\theta}, \quad z = 1 + \epsilon e^{-i\phi}. \tag{C.1}$$

The amplitudes vanish in the limit $r \to \infty$ when the angles $\theta$ and $\phi$ are inside the region plotted in Figure 10. We see that we can indeed find continuous deformations of the integration contours which are always inside the convergent regions.

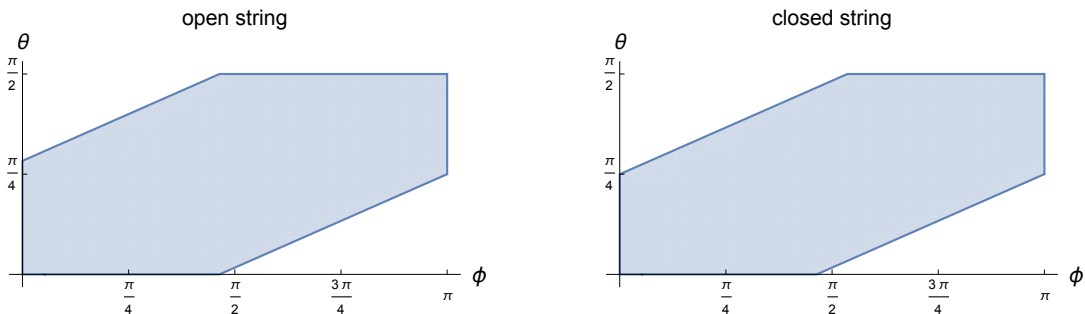

**Figure 10**. The convergence region $\Theta^{12\leftrightarrow34}(z)$ of the open and closed string amplitudes for $\epsilon = 10^{-6}$.

## D   Conformal partial wave representation

The four-point celestial amplitude (2.12) can be expanded in the conformal partial wave basis as

$$f_{\Delta_i,\ell_i}(z,\bar{z}) = \sum_{\ell=-\infty}^{\infty} \int_0^{\infty} \frac{d\nu}{2\pi} \frac{p_\ell(\Delta_i,\ell_i;\nu)}{n_\ell(\nu)} \Psi^{\text{conf.}}_{\Delta_i,\ell_i;1+i\nu,\ell}(z,\bar{z}), \tag{D.1}$$

where $\Psi^{\text{conf.}}_{\Delta_i,\ell_i;1+i\nu,\ell}(z,\bar{z})$ is the conformal partial wave normalized such that

$$\frac{1}{2}\int \frac{d^2z}{|z|^4} \Psi^{\text{conf.}}_{\Delta_i,\ell_i;1+i\nu,\ell}(z,\bar{z})\Psi^{\text{conf.}}_{1-\Delta_i,-\ell_i;1-i\nu',-\ell'}(z,\bar{z}) = n_\ell(\nu) \times 2\pi\delta_{\ell,\ell'}\delta(\nu-\nu'), \tag{D.2}$$

for external dimensions in the principal series, i.e. $\Delta_i \in 1 + i\mathbb{R}$. The normalization $n_\ell(\nu)$ is[12]

$$n_\ell(\nu) = \frac{2\pi^3}{\ell^2 + \nu^2}. \tag{D.3}$$

---

[12]We follow the convention in [32, 33].

For the scalar amplitude with the tree-level massive scalar exchange in the $12 \leftrightarrow 34$ kinematics, the imaginary part of the expansion coefficient $p_\ell(\Delta_i; \nu)$ factorizes as [6]

$$\mathbf{Im}\, p_\ell(\Delta_i; \nu) = \delta_{\ell,0} \pi^2 m^2 C_{\Delta_1,\Delta_2;1+i\nu} C_{\Delta_3,\Delta_4;1-i\nu}\,, \tag{D.4}$$

where $C_{\Delta_1,\Delta_2;\Delta_3}$ is the 3-point coefficient of the celestial amplitude of two massless and one massive scalars. Using the shadow representation of the conformal partial wave, the factorization (D.4) is equivalent to

$$
\begin{aligned}
\mathbf{Im}\, \tilde{\mathcal{A}}_{\Delta_i}^{12\leftrightarrow 34}(z_i, \bar{z}_i) = {} & \int_0^\infty \frac{\mathrm{d}\nu}{2\pi} \frac{\pi^2 m^2}{n_\ell(\nu)} \int \mathrm{d}^2 z \\
& \times\ \tilde{\mathcal{A}}_{3,\Delta_1,\Delta_2;1+i\nu}(z_1, \bar{z}_1, z_2, \bar{z}_2, z, \bar{z}) \tilde{\mathcal{A}}_{3,\Delta_3,\Delta_4;1-i\nu}(z_3, \bar{z}_3, z_4, \bar{z}_4, z, \bar{z})\,.
\end{aligned} \tag{D.5}
$$

where $\tilde{\mathcal{A}}_{\Delta_1,\Delta_2;\Delta_3}(z_i, \bar{z}_i)$ is the three-point celestial amplitude,

$$\tilde{\mathcal{A}}_{\Delta_1,\Delta_2;\Delta_3}(z_i, \bar{z}_i) = \frac{C_{\Delta_1,\Delta_2;\Delta_3}}{|z_{12}|^{\Delta_1+\Delta_2-\Delta_3}|z_{13}|^{\Delta_1+\Delta_3-\Delta_2}|z_{23}|^{\Delta_2+\Delta_3-\Delta_1}}\,. \tag{D.6}$$

In this appendix, we generalize the above result to the four-point scalar celestial amplitude with the tree-level massive spin-$J$ particle exchange. We show that the imaginary part of such an amplitude factorizes as an integral of a product of two three-point celestial amplitudes of two massless scalars and one massive spin-$J$ particle,

$$
\begin{aligned}
\mathbf{Im}\, \tilde{\mathcal{A}}_{\Delta_i,J}^{12\leftrightarrow 34}(z_i, \bar{z}_i) = {} & \pi^2 m^{2J+2} J! \sum_{\ell=-J}^{J} \int_0^\infty \frac{\mathrm{d}\nu}{2\pi} \frac{\mu_{J,\ell}(\nu)}{2^{|\ell|} n_\ell(\nu)} \int \mathrm{d}^2 z \\
& \times\ \tilde{\mathcal{A}}_{3,\Delta_1,\Delta_2;1+i\nu,\ell}^{(J)}(z_1, \bar{z}_1, z_2, \bar{z}_2, z, \bar{z}) \tilde{\mathcal{A}}_{3,\Delta_3,\Delta_4;1-i\nu,-\ell}^{(J)}(z_3, \bar{z}_3, z_4, \bar{z}_4, z, \bar{z})\,,
\end{aligned} \tag{D.7}
$$

where $\mu_{J,\ell}(\nu)$ is

$$\mu_{J,\ell}(\nu) = (-1)^{|\ell|} \frac{2^{J-|\ell|}(|\ell|+1)_{J-|\ell|}(|\ell|+\frac{1}{2})_{J-|\ell|}}{(J-|\ell|)!(2|\ell|+1)_{J-|\ell|}}\,. \tag{D.8}$$

Let us first focus on the right hand side of (D.7). The 3-point celestial amplitude is given by the integral [34]

$$
\begin{aligned}
\tilde{\mathcal{A}}_{3,\Delta_i,\Delta,\ell}^{(J)}(z_i, \bar{z}_i) = {} & \left( \prod_{i=1}^{2} \int_0^\infty d\omega_i\, \omega_i^{\Delta_i-1} \right) \int \frac{dy}{y^3} dz' d\bar{z}' \\
& \times \sum_{b=-J}^{J} G_{\ell,b}^{(J)}(\hat{k}; z, \bar{z}) A_{3,b}(p_1, p_2, k) \delta^4(k+p_1+p_2)\,,
\end{aligned} \tag{D.9}
$$

where $A_{3,b}(p_1, p_2, k)$ is the three-point amplitude explicitly given by

$$A_{3,b}(p_1, p_2, k) = p_{12}^{\mu_1} \dots p_{12}^{\mu_J} \epsilon_{b,\mu_1\mu_2\dots\mu_J}\,, \tag{D.10}$$

with all momenta outgoing. $\epsilon_{b,\mu_1\mu_2\dots\mu_J}$ is the polarization tensor for spin-$s$ particle. The massless momenta $p_1$, $p_2$ are parametrized as before by $p_i = -\omega_i q_i$ and (2.1). The massive momentum $k$ is parametrized by

$$k = m\hat{k}\,, \quad \hat{k}^\mu = \frac{1}{2y}(1+y^2+|z'|^2, 2\mathrm{Re}(z'), 2\mathrm{Im}(z'), 1-y^2-|z'|^2)\,. \tag{D.11}$$

$G_{\ell,b}^{(J)}(\hat{k}; z, \bar{z})$ is the integration weight matrix that relates the spin-$J$ massive irreducible representations of the little group on $\mathbb{R}^{1,3}$ to the spin-$\ell$ representations of the conformal group on the celestial sphere. The integration weight matrix $G_{\ell,b}^{(J)}(\hat{k}; z, \bar{z})$ is computed in [34]

$$\sum_{b=-J}^{J} G_{\ell,b}^{(J)}(\hat{k}; z, \bar{z}) \epsilon_b^{\mu_1 \mu_2 \cdots \mu_J} = \frac{1}{J! \left(\frac{1}{2}\right)_J} K^{\mu_1} K^{\mu_2} \cdots K^{\mu_J} G_{\ell,\Delta}^{(J)}(\hat{k}, Y; z, \bar{z})\Big|_{Y=0},$$

$$G_{\ell,\Delta}^{(J)}(\hat{k}, Y; z, \bar{z}) = \begin{cases} \frac{(Y \cdot q)^{J-\ell}\left[(\hat{k} \cdot q)(Y \cdot \partial_{\bar{z}} q) - (\hat{k} \cdot \partial_{\bar{z}} q)(Y \cdot q)\right]^{\ell}}{(-\hat{k} \cdot q)^{\Delta+J}} & \text{for} \quad \ell \geq 0, \\ \frac{(Y \cdot q)^{J+\ell}\left[(\hat{k} \cdot \partial_z q)(Y \cdot q) - (\hat{k} \cdot q)(Y \cdot \partial_z q)\right]^{-\ell}}{(-\hat{k} \cdot q)^{\Delta+J}} & \text{for} \quad \ell < 0, \end{cases} \tag{D.12}$$

$$K_\mu(\hat{k}, Y) = \frac{1}{2}\left[\partial_{Y^\mu} + \hat{k}_\mu(\hat{k} \cdot \partial_Y)\right] + (Y \cdot \partial_Y)\partial_{Y^\mu}$$
$$+ \hat{k}_\mu(Y \cdot \partial_Y)(\hat{k} \cdot \partial_Y) - \frac{1}{2} Y_\mu \left(\partial_Y^2 + (\hat{k} \cdot \partial_Y)^2\right),$$

where the null vector $q$ is parametrized by $z$ and $\bar{z}$ as in (2.1). Plugging this formula into (D.10), we find

$$\tilde{\mathcal{A}}_{3,\Delta_i,\Delta,\ell}^{(J)}(z_i, \bar{z}_i) = \frac{1}{J!(\frac{1}{2})_J} \left(\prod_{i=1}^{2} \int_0^\infty d\omega_i\, \omega_i^{\Delta_i-1}\right) \int \frac{dy}{y^3} dz' d\bar{z}'$$
$$\times \delta^4(p + p_1 + p_2)(p_{12} \cdot K)^J G_{\ell,\Delta}^{(J)}(\hat{k}, Y; z, \bar{z}). \tag{D.13}$$

Now, the right hand side of (D.7) can be simplified as

$$\frac{\pi^2 m^{2J+2} J!}{\left(J!(\frac{1}{2})_J\right)^2} \left(\prod_{i=1}^{4} \int_0^\infty d\omega_i\, \omega_i^{\Delta_i-1}\right) \sum_{\ell=-J}^{J} \int_0^\infty \frac{d\nu}{2\pi} \frac{\mu_{J,\ell}(\nu)}{2^{|\ell|} n_\ell(\nu)} \left(\prod_{i=1}^{2} \int \frac{dy_i}{y_i^3} d^2 z_i'\right) \int d^2 z$$
$$\times \delta^{(4)}(p_1 + p_2 + k_1)\delta^{(4)}(p_3 + p_4 - k_2)$$
$$\times \left[\left(p_{12} \cdot K(\hat{k}_1, Y_1)\right)^J G_{\ell,1+i\nu}^{(J)}(\hat{k}_1, Y_1; z, \bar{z})\right] \left[\left(p_{34} \cdot K(\hat{k}_2, Y_2)\right)^J G_{-\ell,1-i\nu}^{(J)}(\hat{k}_2, Y_2; z, \bar{z})\right]\Big|_{Y_1=Y_2=0}$$
$$= \frac{\pi^2 m^{2J+2} J!}{4\pi \left(J!(\frac{1}{2})_J\right)^2} \left(\prod_{i=1}^{4} \int_0^\infty d\omega_i\, \omega_i^{\Delta_i-1}\right) \int \frac{dy_1}{y_1^3} d^2 z_1' \delta^{(4)}(p_1 + p_2 + k_1)\delta^{(4)}(p_3 + p_4 - k_1)$$
$$\times \left(p_{12} \cdot K(\hat{k}_1, Y_1)\right)^J \left(p_{34} \cdot K(\hat{k}_1, Y_2)\right)^J (Y_1 \cdot Y_2)^J\Big|_{Y_1=Y_2=0}$$
$$= \pi m^{2J} \left(\prod_{i=1}^{4} \int_0^\infty d\omega_i\, \omega_i^{\Delta_i-1}\right) \delta^{(4)}(p_1 + p_2 + p_3 + p_4)\delta(s - m^2) P_J\left(\frac{u-t}{m^2}\right), \tag{D.14}$$

where in the second equality, we have used the orthogonality condition [34]

$$\sum_{\ell=-J}^{J} \int_{-\infty}^\infty d\nu \frac{\mu_{J,\ell}(\nu)}{2^{|\ell|} n_\ell(\nu)} \int d^2 z\, G_{\ell,1+i\nu}^{(J)}(\hat{k}_1, Y_1; z, \bar{z}) G_{-\ell,1-i\nu}^{(J)}(\hat{k}_2, Y_2; z, \bar{z}) = \delta(\hat{k}_1, \hat{k}_2)(Y_1 \cdot Y_2)^s, \tag{D.15}$$

where the delta function $\delta(\hat{k}_1, \hat{k}_2)$ is defined by

$$\int_0^\infty \frac{dy_2}{y_2^3} \int d^2 z_2' \delta(\hat{k}_1, \hat{k}_2) F(\hat{k}_2) = F(\hat{k}_1). \tag{D.16}$$

In the third equality of (D.14), we have used the identities

$$\int \frac{\mathrm{d}y_1}{y_1^3} \mathrm{d}^2 z_1' \delta^{(4)}(p_1 + p_2 + k_1) = \frac{4}{m^2} \delta(s - m^2),$$

$$\left(p_{12} \cdot K(\hat{k}_1, Y_1)\right)^J \left(p_{34} \cdot K(\hat{k}_1, Y_2)\right)^J (Y_1 \cdot Y_2)^J \Big|_{Y_1 = Y_2 = 0} = J! \left(\frac{(2J-1)!!}{2^J}\right)^2 P_J\left(\frac{u-t}{m^2}\right).$$

(D.17)

Next, let us look at the left hand side of (D.7). The four-point celestial amplitude can be computed using the Mellin integral (2.2) and the formula (2.15) with $\ell_i = 0$,

$$\mathbf{Im}\, \tilde{\mathcal{A}}_{\Delta_i, J}^{12 \leftrightarrow 34}(z_i, \bar{z}_i) = \left(\prod_{i=1}^{4} \int_0^\infty d\omega_i\, \omega_i^{\Delta_i - 1}\right) \delta^{(4)}(p_1 + \cdots + p_4)$$

$$\times m^{2J} \mathbf{Im} \left[-\frac{P_J\left(\frac{u-t}{m^2}\right)}{s - m^2 + i\epsilon} - \frac{P_J\left(\frac{s-t}{m^2}\right)}{u - m^2 + i\epsilon} - \frac{P_J\left(\frac{u-s}{m^2}\right)}{t - m^2 + i\epsilon}\right]$$

$$= \pi m^{2J} \left(\prod_{i=1}^{4} \int_0^\infty d\omega_i\, \omega_i^{\Delta_i - 1}\right) P_J\left(\frac{u-t}{m^2}\right) \delta(s - m^2) \delta^{(4)}(p_1 + \cdots + p_4).$$

(D.18)

(D.14) and (D.18) matches exactly; hence, the factorization formula (D.7) follows.

By the conformal symmetry, the three-point celestial amplitude $\tilde{\mathcal{A}}_{3,\Delta_1,\Delta_2;\Delta,\ell}^{(J)}(z_i, \bar{z}_i)$ takes the form as

$$\tilde{\mathcal{A}}_{3,\Delta_1,\Delta_2;\Delta,\ell}^{(J)}(z_i, \bar{z}_i) = \frac{C_{\Delta_1,\Delta_2;\Delta,\ell}^{(J)}}{|z_{12}|^{\Delta_1+\Delta_2-(\Delta-\ell)}|z_{13}|^{\Delta_1+(\Delta-\ell)-\Delta_2}|z_{23}|^{\Delta_2+(\Delta-\ell)-\Delta_1}} \left(\frac{z_{12}}{z_{13}z_{23}}\right)^\ell.$$

(D.19)

Plugging this into (D.7) and using the form (2.12) of the four-point celestial amplitude, we find

$$\mathbf{Im}\, f_{\Delta_i, J}^{12 \leftrightarrow 34}(z, \bar{z}) = \pi^2 m^{2J+2} J! \sum_{\ell=-J}^{J} \int_0^\infty \frac{\mathrm{d}\nu}{2\pi} \frac{\mu_{J,\ell}(\nu)}{2^{|\ell|} n_\ell(\nu)} C_{\Delta_1,\Delta_2;1+i\nu,\ell}^J C_{\Delta_3,\Delta_4;1-i\nu,-\ell}^{-J} \Psi_{\Delta_i,\ell_i;1+i\nu,\ell}^{\text{conf.}}(z, \bar{z}).$$

(D.20)

Note that with the prefactor (2.14), the left hand side of (D.20) is exactly the scalar Poincare partial wave (3.23), i.e.

$$\mathbf{Im}\, f_{\Delta_i, \ell}^{12 \leftrightarrow 34}(z, \bar{z}) = (z-1)^{\frac{\Delta_1-\Delta_2-\Delta_3+\Delta_4}{2}} \delta(iz - i\bar{z}) \Psi_{m,l}^{12 \leftrightarrow 34}(\mathbf{\Delta}, z).$$

(D.21)

Thus, (D.20) gives a conformal partial wave representation of the scalar Poincaré partial wave!

Finally, in [34], the three-point coefficients $C_{\Delta_1,\Delta_2;\Delta_3}^J$ for $J = 0, 1, 2$ are computed, and recursion relations for the general three-point coefficients are derived. In Appendix E, we compute the general three-point coefficients by directly evaluating the Mellin integral (D.13).

# E  Computation of $C^{(J)}_{\Delta_1,\Delta_2;\Delta,\ell}$

Let us compute the structure constant $C^{(J)}_{\Delta_1,\Delta_2;\Delta,\ell}$ by explicitly working out the Mellin integral (D.13). The momentum conservation delta function gives

$$y = \frac{m}{2(\omega_1 + \omega_2)}\,, \quad z' = \frac{z_1\omega_1 + z_2\omega_2}{\omega_1 + \omega_2}\,, \quad \bar{z}' = \frac{\bar{z}_1\omega_1 + \bar{z}_2\omega_2}{\omega_1 + \omega_2}\,, \quad \omega_1\omega_2 = \frac{m^2}{4|z_{12}|^2}\,, \tag{E.1}$$

and the Jacobian

$$|\text{Jacobian}| = \frac{8m|z_{12}|^4\omega_2^2}{(m^2 + 4|z_{12}|^2\omega_2^2)^3}\,. \tag{E.2}$$

The inner products that appear in the $(p_{12} \cdot K)^J G^{(J)}_{\ell,\Delta}(\hat{k}, Y; z, \bar{z})$ are summarized as

$$2p_{12} \cdot q = -4|z - z_2|^2\omega_2 + \frac{m^2|z - z_1|^2}{|z_{12}|^2\omega_2}\,, \quad 2k \cdot q = -4|z - z_2|^2\omega_2 - \frac{m^2|z - z_1|^2}{|z_{12}|^2\omega_2}\,,$$

$$p_{12}^2 = m^2\,, \quad k \cdot p_{12} = 0\,, \tag{E.3}$$

$$(k \cdot q)(p_{12} \cdot \partial_{\bar{z}}q) - (k \cdot \partial_{\bar{z}}q)(q \cdot p_{12}) = -\frac{2m(z - z_1)(z - z_2)}{z_{12}}\,.$$

We find

$$p_{12} \cdot K = \left(N_Y + \frac{1}{2}\right)p_{12} \cdot \partial_Y - \frac{1}{2}(p_{12} \cdot Y)\left(\partial_Y^2 + (\hat{k} \cdot \partial_Y)^2\right)\,, \tag{E.4}$$

where $N_Y = Y \cdot \partial_Y$ simply counts the number of $Y$. We have the following identities

$$\partial_Y^2 G^{(J)}_{\ell,\Delta}(\hat{k}, Y; z, \bar{z}) = 0\,, $$
$$(\hat{k} \cdot \partial_Y)^2 G^{(J)}_{\ell,\Delta}(\hat{k}, Y; z, \bar{z}) = (J - |\ell|)(J - |\ell| - 1)G^{(J-2)}_{\ell,\Delta}(\hat{k}, Y; z, \bar{z})\,. \tag{E.5}$$

Using these formulae, we find

$$(p_{12} \cdot K)^J G^{(J)}_{\ell,\Delta}(\hat{k}, Y; z, \bar{z})\Big|_{Y=0}$$

$$= \frac{1}{2^J} \sum_{n=0}^{\frac{J}{2}} (-1)^n (2J - 1 - 2n)!!(2n - 1)!!\binom{J}{2n}$$

$$\times m^{2n}(p_{12} \cdot \partial_Y)^{J-2n}(\hat{k}_{12} \cdot \partial_Y)^{2n}G^{(J)}_{\ell,\Delta}(\hat{k}, Y; z, \bar{z})\Big|_{Y=0} \tag{E.6}$$

$$= \frac{1}{2^J} \sum_{n=0}^{\frac{J}{2}} (-1)^n (2J - 1 - 2n)!!(2n - 1)!!\binom{J}{2n}$$

$$\times m^{2n}\frac{(J - |\ell|)!(J - 2n)!}{(J - |\ell| - 2n)!}G^{(J-2n)}_{\ell,\Delta}(\hat{k}, p_{12}, z, \bar{z})\,.$$

Let us assume $\ell \geq 0$, and consider the integral

$$\left(\prod_{i=1}^{2} \int_0^\infty d\omega_i\, \omega_i^{\Delta_i - 1}\right) \int \frac{dy}{y^3} dz' d\bar{z}' \delta^4(p + p_1 + p_2) G_{\ell,\Delta}^{(J-2n)}(\hat{k}, p_{12}; z, \bar{z})$$

$$= 2^{2-\Delta_1-\Delta_2-\Delta} m^{\Delta_1+\Delta_2-4+J-2n} |z_{12}|^{-2\Delta_1} \left(\frac{(z-z_1)(z-z_2)}{z_{12}}\right)^\ell \frac{1}{|z-z_2|^{2(\Delta+\ell)}}$$

$$\times \int_0^\infty d\omega_2 \sum_{p=0}^{J-2n-\ell} (-1)^{\ell+p} \binom{J-2n-\ell}{p} \frac{\omega_2^{\Delta_2-\Delta_1-1+\Delta+\ell+2p}\left(\frac{|z-z_1|^2}{|z_{12}|^2|z-z_2|^2}\right)^{J-2n-\ell-p}}{\left(\omega_2^2 + \frac{|z-z_1|^2}{|z_{12}|^2|z-z_2|^2}\right)^{\Delta+J-2n}}$$

$$= 2^{1-\Delta_1-\Delta_2-\Delta} m^{\Delta_1+\Delta_2-4+J-2n} \sum_{p=0}^{J-2n-\ell} (-1)^{\ell+p} \binom{J-2n-\ell}{p}$$

$$\times B\left(\frac{\Delta+\ell+2p-\Delta_1+\Delta_2}{2}, \frac{\Delta-\ell+2J-4n-2p+\Delta_1-\Delta_2}{2}\right)$$

$$\times \left(\frac{(z-z_1)(z-z_2)}{z_{12}}\right)^\ell |z-z_1|^{\Delta_2-\Delta_1-\Delta-\ell} |z-z_2|^{\Delta_1-\Delta_2-\Delta-\ell} |z_{12}|^{\Delta+\ell-\Delta_1-\Delta_2}.$$

$$\text{(E.7)}$$

The three-point coefficient is

$$C_{\Delta_1,\Delta_2;\Delta,\ell}^{(J)} = \frac{m^{\Delta_1+\Delta_2-4+J}}{2^{\Delta_1+\Delta_2+\Delta+J-1} J! (\frac{1}{2})_J} \sum_{n=0}^{\frac{J}{2}} \sum_{p=0}^{J-2n-\ell} (-1)^{\ell+n+p} \binom{J}{2n}\binom{J-2n-\ell}{p}$$

$$\times \frac{(J-|\ell|)!(J-2n)!(2J-1-2n)!!(2n-1)!!}{(J-|\ell|-2n)!}$$

$$\times B\left(\frac{\Delta+\ell+2p-\Delta_1+\Delta_2}{2}, \frac{\Delta-\ell+2J-4n-2p+\Delta_1-\Delta_2}{2}\right).$$

$$\text{(E.8)}$$

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
