# Peer review of "Bulk locality from the celestial amplitude"

_SciPost Physics_

## Round 1 · Referee Report · Anonymous (Referee 1) · 2021-11-18

Strengths
-
The authors carefully study the analytic properties of the celestial amplitudes. The implication of bulk locality is also emphasized.
-
In celestial CFT, one of the most puzzling aspects is crossing symmetry. The authors perform a thorough analysis of the relations between celestial correlators computed from different crossing chanels.
-
The authors demonstrated the general statements in many examples, including scalar field theory and string theory.
Weaknesses
- While the authors uncover and demonstrate several interesting properties of the celestial amplitudes, the physical interpretation for some of the properties is not clear.
Report
Requested changes
-
Above (2.6), there seems to be a missing reference.
-
It seems to me that when the authors discussed the imaginary part of the celestial amplitude, they assumed $\beta$ to be real. It is not entirely clear to me where and why did the authors make this assumption. Perhaps it's around (2.31), but the convergence of the Mellin integral seems to allow for an imaginary part of $\beta$.
-
There seems to be a typo in the $z(1-z)$ factor in the first line of (5.1).
-
The analytic property (5.5) is very interesting. Can the authors provide some physical interpretation? Should we view (5.5) as the crossing symmetry for the celestial amplitudes?
Author: Chi-Ming Chang on 2022-01-28 [id 2125]
(in reply to Report 1 on 2021-11-18)We thank the referee for the comments, and will resubmit to address referee's points.

---

## Round 2 · Referee Report · Anonymous · 2022-2-27

Report

The celestial holography program aims to construct a 2D holographic theory, a Celestial Conformal Field Theory (CCFT), that computes scattering amplitudes for 4D quantum gravity in asymptotically flat space as 2D conformal correlators (also known as celestial amplitudes) on the celestial sphere at null infinity. So far a successful example of CCFT has not been constructed yet, and it is an active area of research to understand the properties of such a theory. Other than matching symmetries on the bulk and boundary sides, there is more information about the bulk physics encoded in the celestial amplitudes. Inspired by the parallel development of the S-matrix bootstrap program, one is led to ask: how are bulk unitarity and locality reflected on the analytic structures of celestial amplitudes? This work takes the initiative to explore this direction, with the focus on 4-point massless celestial amplitudes. Here I would like to highlight a few results that I believe will be useful for this research program:

1. The formula (3.5) for the imaginary part of the celestial amplitudes of 4-point massless particles.

2. Positivity of the imaginary part of the celestial amplitude expanded in Poincare partial waves.

3. The discussion of the analytic continuation of the celestial amplitudes in the cross ratio $z$.

4. Dispersion relation (6.4) of celestial amplitudes.

These results are novel, and are checked analytically and numerically with massless scalars and string theory examples. This work will help build a bootstrap approach to celestial amplitudes. The present manuscript is reasonably well-written, and the technical arguments are, for the most part, presented clearly and easy to follow. I would like to make a few comments and recommend some clarifications (see below). Once these are addressed, I would be happy to recommend the present manuscript for publication in SciPost.

Requested changes

1-Before (2.18) it says ``celestial scalar amplitudes", so it is confusing that equations (2.18), (2.19), (2.23) have helicity ($\ell_i$) dependence.

2-After (2.25), in $s$-kinematics we should have $z\geq 1$.

3-I believe the result (2.33) was first obtained in reference [22] and therefore they should be mentioned.

4-It should be mentioned that (3.6) is for scalars.

5-Even though it is somewhat clear from the context, $m,m_i,J,J_i$ are not defined in section 3.2. Also, it seems that there should be a sum over $J_i$ in (3.7) and (3.10). The left hand side of these expressions are certainly not functions of $J$'s.

6-It seems that (3.24) only captures the contributions from the factorization poles (labeled by $a$). It should be clarified. Also, there should be a sum over $J_a$.

7-Related to the last point, I wonder if the authors have any comments on the positivity for the part associated with the branch cut in (3.5).

8-Footnote 6 suggests that the authors have worked out the case for odd spins. I recommend that the more general expressions for all spins should be included.

9-Is (5.5) only true for 4 external scalars, or is it true for any massless particles of any spins?

10-It would be great if the authors can comment on the motivations/justifications for the two assumptions after (6.7).

  • validity: high
  • significance: high
  • originality: high
  • clarity: good
  • formatting: -
  • grammar: good

Author:  Chi-Ming Chang  on 2022-03-12  [id 2285]

(in reply to Report 1 on 2022-02-27)

We thank the referee for the comments. We will resubmit the paper with the changes listed below to address referee's points.

1.The “scalar” has been removed. 2. Typo fixed. 3. Reference added. 4. A sentence added above (3.6). 5-7. Sentences addressing these two points are added around (3.8). 8-9. Original footnote 6 removed. Sentences added below (5.5). 10. Explanations added below the analyticity assumption. The boundedness condition is moved to Section 3.1 with some justifications added below it.

---

## Round 2 · Referee Report · Anonymous · 2022-3-8

Report

The authors have addressed all the comments in my previous report. It is ready for publication in my opinion.

---

## Round 2 · List of Changes

1. The “[]” above (2.6) is removed.
2. A typo in (5.1) is corrected.
3. Sentences added around (5.5) to address referee's comment.

---

## Editorial Decision

resubmitted